# Near Optimal Best Arm Identification for Clustered Bandits

**Yash** [1]  **Avishek Ghosh** [2]  **Nikhil Karamchandani** [1]

## Abstract

This work investigates the problem of best arm identification for multi-agent multi-armed bandits. We consider $N$ agents grouped into $M$ clusters, where each cluster solves a stochastic bandit problem. The mapping between agents and bandits is *a priori* unknown. Each bandit is associated with $K$ arms, and the goal is to identify the best arm for each agent under a $\delta$-probably correct ($\delta$-PC) framework, while minimizing sample complexity and communication overhead. We propose two novel algorithms: *Clustering then Best Arm Identification* (`Cl-BAI`) and *Best Arm Identification then Clustering* (`BAI-Cl`). `Cl-BAI` employs a two-phase approach that first clusters agents based on the bandit problems they are learning, followed by identifying the best arm for each cluster. `BAI-Cl` reverses the sequence by identifying the best arms first and then clustering agents accordingly. Both algorithms exploit the successive elimination framework to ensure computational efficiency and high accuracy. Theoretical analysis establishes $\delta$-PC guarantees for both methods, derives bounds on their sample complexity, and provides a lower bound for the problem class. Moreover, when $M$ is small (a constant), we show that the sample complexity of (a variant of) `BAI-Cl` is (order-wise) minimax optimal. Experiments on synthetic and real-world (Movie Lens, Yelp) data demonstrates the superior performance of the proposed algorithms in terms of sample and communication efficiency, particularly in settings where $M \ll N$.

[1]Department of Electrical Engineering, IIT Bombay, India [2]Department of Computer Science and Engineering, IIT Bombay, India. Correspondence to: Avishek Ghosh <avishek_ghosh@iitb.ac.in>.

*Proceedings of the 42nd International Conference on Machine Learning*, Vancouver, Canada. PMLR 267, 2025. Copyright 2025 by the author(s).

## 1. Introduction

Multi armed bandits (MAB) (1) has become a classical framework for modeling sequential learning as it carefully captures the exploration-exploitation dilemma. It has shown great success in applications like advertisement (Ad) placement, clinical trials, and recommendation system (see (1; 2)). In the past decade, there has been an enormous increase in the amount of processed data to the extent that it has become pivotal to distribute the learning process and leverage collaboration among multiple agents.

In lieu of this, (3) introduced Federated Multi Armed Bandit (F-MAB), where we have $N$ agents and a central learner; the agents can only talk to one another through the central learner. This framework is particularly interesting when the (sequential) data is observed in a distributed fashion and the action is also taken by the agents individually. This is a highly decentralized paradigm with lots of challenges (see (4; 5; 6)). In this paper, we propose and analyze learning algorithms that aim to address one of the major challenges in F-MAB–heterogeneity across agents.

In F-MAB, the problem of heterogeneity naturally arises since the preferences of different agents may not be identical. In the movie recommendation example, different agents prefer different genres of movies like comedy, drama, action etc. Hence, the recommendation platform needs to identify agents based on their preferences and suggest movies accordingly. A similar situation pops up in Ad placement, where the type of Ads shown to different people might be based on their taste. Moreover, in social recommendation platforms like Yelp, this heterogeneity effect needs to be addressed for better restaurant recommendation.

In this work, we model the heterogeneity of the agents through clustering. Note that clustering is a canonical way to group *similar* agents for better collaboration. We consider a multi-agent multi-armed bandit problem with $N$ agents, grouped into $M$ clusters (which are *a priori* unknown). All the agents have access to a common collection of $K$ arms, i.e., the set of arms is common for all $N$ agents across the $M$ clusters. Each cluster $m \in [M]$[1] is trying to learn a stochastic bandit problem with best arm $k_m^*$. Hence, all the agents belonging to cluster $m$ share the (unique) best arm $k_m^*$, and

[1]For a positive $r$, we denote $[r] = \{1, 2, ..., r\}$

agents belonging to different clusters will have different best arms, i.e., if agents $i_1$ and $i_2$ belong in clusters $m_1$ and $m_2$ (with $m_1 \neq m_2$), we have $k_{m_1}^* \neq k_{m_2}^*$.

Federated Bandits (F-MAB) has received a lot of interest in the past few years. In (7; 8; 9), the authors consider distributed pure exploration with $N$ agents learning the same bandit problem, with some communication allowed amongst them and study the tradeoff between the number of arm pulls needed per agent and the number of rounds of communication. Moreover, in (3; 10; 11) the *federated pure exploration* setting is studied where multiple agents are learning different bandit problems (i.e., each agent has its own associated mean reward vector for the arms) and need to communicate with a central server to learn the arm with the highest sum of mean rewards across the agents. The setting of (12) is similar to above works (i.e., unstructured with no clustering), where the goal is to find not just the global best arm, but also the local best arms for each agent in a communication efficient manner. Furthermore, (3; 13) consider the federated bandits setup within a regret minimization framework.

In this paper, we address the problem of *best arm identification* (BAI) for all $N$ agents in F-MAB in a heterogeneous (clustered) setup. We propose and analyze *Successive Elimination* based learning algorithms for this task. Our algorithms are efficient in terms of sample complexity (the number of pulls) as well as communication cost between the agents and the central leaner (which is desirable in F-MAB, see (12)).

Clustering in F-MAB also has a rich literature. In (14; 15; 16; 17; 18; 19) the authors study regret minimization in a clustered linear bandits framework. Also (20) studies regret minimization where agents are divided into clusters and agents in the same cluster have the same mean rewards vector. The paper employs techniques from online matrix completion under an incoherence condition. Another line of works (see (21; 22; 23)) assumes that each arm pull generates a vector feedback and arms in the same cluster have the same mean reward vector.

Perhaps the work closest to us is (24). Here, agents are grouped into roughly equal sized clusters and all agents in one cluster are solving the same bandit, with the goal being to minimize an appropriately defined cumulative group regret. It is assumed that agents form a graph and can talk to one another through a *gossip* style protocol. Although the cluster structure here is similar to ours, (24) studies group regret whereas we focus on the sample complexity for best arm identification. Moreover, (24) allows gossip style communication protocol which is prohibited in our F-MAB setup. Finally, (25) uses a similar setting and identifies the best arm for a single bandit instance by several agents, each of which can only access a subset of the arms, thus necessitating communication.

## 1.1. Our Contributions

*Algorithm Design:* We propose and analyze two novel algorithms; (i) *Clustering then Best Arm Identification* (`Cl-BAI`) and (ii) *Best Arm Identification then Clustering* (`BAI-Cl`). We use *successive elimination* for clustering and BAI primarily because of its simplicity and easy-tuning ability. We remark that other algorithms may also be used for these in our framework. Both the algorithms judiciously pull arms so that both clustering and BAI can be done in a sample efficient manner. Our algorithms are also efficient in terms of communication cost (to be defined shortly) between the agents and the central learner.

*Theoretical Guarantees:* For a fixed confidence $\delta \in (0,1)$, we obtain the sample complexity for `CL-BAI` and `BAI-CL` for identifying the best arm for all $N$ agents. Leveraging a separability condition (for identifiability) across clusters, we analyze `CL-BAI`. On the other hand, for `BAI-CL`, using a probabilistic argument, similar to the classical coupon collector problem, we first identify representatives from each cluster, and then judiciously construct a subset of candidate best arms to reduce the number of arm-pulls. We characterize the benefits of `BAI-CL` over `CL-BAI` rigorously. We also study a variation of `BAI-CL`, namely `BAI-CL++`.

*Lower Bound and Optimality:* Considering a large class of problem instances and using change-of-measure style arguments, we obtain a minimax lower bound over the class of *all* learning algorithms. An interesting feature of our lower bound is that it is the maximum of two terms, each corresponding to a different sub-task which any feasible scheme should be able to complete; one being identifying the set of best arms across the $M$ bandits and the second being identifying for each agent the index of the bandit problem that it is learning. Finally, we show that if the number of clusters, $M$, is small (constant), the algorithm `BAI-CL++` is order-wise minimax optimal in terms of sample complexity.

*Experiments:* We validate the theoretical findings through extensive experiments, both on synthetic and real-world datasets. We find that our proposed schemes are able to efficiently cluster and significantly reduce the overall sample complexity. For example, in a movie recommendation application with 100 users, clustered into 6 different age groups, each with different preferences derived from the MovieLens-1M dataset, `BAI-Cl++` is able to provide a 72% improvement in the sample complexity over a naive cluster-oblivious scheme. A 65% improvement is observed in a similar experiment conducted with the Yelp dataset.

## 2. Problem Setup

We have $N$ agents, each of which is trying to learn one out of $M$ stochastic bandit problems. Let $\mathcal{M} : [N] \to [M]$ denote the mapping from the set of agents to the set of bandits

which is not known apriori. Each of the $M$ bandit problems is associated with a common collection of $K$ arms. For each $m \in [M], k \in [K]$, arm $k$ in bandit $m$ is associated with a reward distribution $\Pi_{m,k}$, which we will assume to be 1-subGaussian[2] with a priori unknown mean $\mu_{m,k} \in \mathbb{R}$. Each pull of an arm results in a random reward, drawn independently from the corresponding reward distribution. Furthermore, we define the best arm $k_m^*$ for bandit $m$ as the arm with the largest mean amongst the arms of bandit $m$, i.e., $k_m^* := \arg\max_k \mu_{m,k}$; we assume that $k_m^*$ is unique for each $m$. Finally, for each bandit $m$, we will denote the gap between the mean rewards of the best arm $k_m^*$ and another arm $j$ as $\Delta_{m,j} = \mu_{m,k_m^*} - \mu_{m,j}$; and let $\Delta_{m,k_m^*} = \min_{j \neq k_m^*} \Delta_{m,j}$.

We will make the following assumption throughout regarding the underlying reward distributions of the $M$ bandits.

**Assumption 2.1.** $\exists \eta > 0$ such that for any two different bandits $a, b$, the best arm $k_a^*$ for bandit $a$ performs at least $\eta$ worse under bandit $b$ than the corresponding best arm $k_b^*$, i.e, $\mu_{b,k_b^*} - \mu_{b,k_a^*} \geq \eta, \forall a, b \in [M], a \neq b$.

The assumption above is natural for several settings and encodes a certain form of 'separability' amongst the different bandit problems; in particular, the above assumption implies that each of the $M$ bandit instances have a different best arm and hence $K \geq M$. As we will see later, this assumption enables us to efficiently match agents with the bandit problem they are solving.

Thus, an instance of the our problem is defined by $\mathcal{I} = ([N], [M], [K], \mathcal{M}, \Pi)$, where $\Pi = (\Pi_{m,k}, m \in [M], k \in [K])$. There is a learner whose goal is to identify the best arm $k_{\mathcal{M}(i)}^*$ for each agent $i$. To accomplish this, the learner can use an online algorithm, say $\mathcal{A}$, which at each time can either choose an agent and an arm to sample based on past observations; or decide to stop and output an estimated collection of best arms given by $(O_1, O_2, ..., O_N)$. Given an error threshold $\delta \in (0,1)$, we say that the algorithm $\mathcal{A}$ is $\delta$-probably correct ($\delta$-PC) if, for any underlying problem instance $\mathcal{I}$, the probability that algorithm output is incorrect is at most $\delta$. More formally, denoting the total number of pulls (random) before stopping time of algorithm $\mathcal{A}$ on instance $\mathcal{I}$ by $T_\delta^{\mathcal{I}}(\mathcal{A})$, $\mathcal{A}$ is $\delta$-PC if, for any instance $\mathcal{I}$,

$$\mathbb{P}\left(T_\delta^{\mathcal{I}}(\mathcal{A}) < \infty, \exists i \in [N] \text{ s.t. } O_i \neq k_{\mathcal{M}(i)}^*\right) \leq \delta.$$

We will measure the performance of a $\delta$-PC algorithm $\mathcal{A}$ by its sample complexity $T_\delta^{\mathcal{I}}(\mathcal{A})$. Our goal in this paper is to design $\delta$-PC schemes for our problem whose sample complexity $T_\delta^{\mathcal{I}}(\mathcal{A})$ is as small as possible. Note that $T_\delta^{\mathcal{I}}(\mathcal{A})$ is itself a random quantity, and our results will be in terms of expectation or high probability bounds.

Note that any online algorithm involves communication from the (central) learner to the different agents, as well as vice-versa. In addition to the sample complexity, we also measure the communication complexity of the schemes we propose[3]. To do so, we will assume a cost of $c_r$ units for communicating a real number and $c_b$ units for each bit corresponding to a discrete quantity (i.e., to communicate $x \in \mathcal{X}$ incurs a total cost of $c_b \cdot \lceil \log |\mathcal{X}| \rceil$ units). In general, one would expect $c_r$ to be significantly larger than $c_b$. For example, if in a system each real number is represented using 32 bits, then we will have $c_r = 32 c_b$.

## 3. Related Work

*Best arm identification:* In MAB literature, finding the best arm with probability at least $1 - \delta$ (for a $\delta \in (0,1)$), also known as the pure exploration problem is well studied; for example see (26; 27; 28; 29; 30; 31) and the references therein. Moreover (32; 33; 34; 35; 36; 37) study various variants of pure exploration, such as identifying $k$ out of top $m$ arms; identifying arms with mean rewards above a threshold etc.

*Federated Bandits (F-MAB):* More recently, distributed learning in MAB, also known as Federated Bandits has received a lot of attention as alluded in Section 1. In (7; 8; 9), the authors study the distributed pure exploration problems with some allowed communication among them. On the other hand, (3; 10; 11; 38; 39; 40; 12) study the federated pure exploration problem with heterogeneous reward structure across agents. Moreover, (3; 13; 41) address the F-MAB problem in a regret minimization framework.

*Clustered Federated bandits:* In F-MAB, one of the major challenges is heterogeneity across agents and hence *Clustered F-MAB* is a popular area of research. In the (simple parametric) linear bandit setup, the clustering problem is studied by (14; 16; 17; 18; 42). Moreover (24; 25; 20; 21; 22) study clustered F-MAB without linear structure, and a detailed discussion as well as comparison of these works with our work are presented in Section 1.

## 4. Algorithm I: `Cl-BAI`

Throughout the algorithm, we will often calculate the average of the reward samples observed from the arm thus far. For a generic arm $k$, we will refer to it by $\hat{\mu}_k$. When considering the $k$-th arm of agent $i$, we will refer to it by $\hat{\mu}_k^i$. Note that the true mean reward for this arm is given by $\mu_{\mathcal{M}(i),k}$.

Our first algorithm, Clustering then Best Arm Identification (`Cl-BAI`), is presented in Algorithm 1 and it consists of two phases. The goal of the first phase is to 'cluster' the agents based on the bandit problem that they are learning, so that for

---

[2]A random variable $X$ is $\sigma$-subGaussian if, for any $t > 0$, $\mathbb{P}(|X - \mathbb{E}[X]| > t) \leq 2\exp\left(-t^2/2\sigma^2\right)$.

[3]This is important in F-MAB since it may be directly related to internet bandwidth of the agents which is resource constraint.

**Algorithm 1** Cl-BAI

1: **Input**: $\delta, \eta$; **Initialize:** $Best\_Arm \leftarrow 0_N$
2: **First phase:**
3: **for** $i \in [N]$ **do**
4:     Agent $i$ runs Successive Elimination:
     $S_i, \hat{\mu}^i = SE([K], \gamma = (\frac{\delta}{12NK})^{4/3}, R = \log(17/\eta))$
5:     Agent $i$ communicates $S_i, \hat{\mu}^i$ to learner
6:     **if** $|S_i| = 1$ **then**
7:       $Best\_Arm[i] = S_i, [N] \to [N] \backslash i$
8:     **end if**
9: **end for**
10: Learner constructs graph $\mathcal{G}$ with $[N]$ as set of vertices.
11: **for** $i,j \in [N]$ **do**
12:     Create edge between vertices $i,j$ if $|\hat{\mu}_k^i - \hat{\mu}_k^j| \leq \eta/2$,
     $\forall\, k \in S_i \cup S_j$
13: **end for**
14: Label connected components as $C_1, C_2, ..., C_m$
15: **Second phase:**
16: **for** $i \in [m]$ **do**
17:     Learner selects one agent $a_i$ from $C_i$ and instructs $a_i$
     to run Successive Elimination
18: **end for**
19: **for** $i \in [M]$ **do**
20:     Agent $a_i$ runs Successive Elimination:
21:     $B_i, \hat{\mu}^{a_i} = SE(S_{a_i}, \gamma = \delta/(2M), R = \infty)$
22:     Agent $a_i$ communicates $B_i$ to learner
23: **end for**
24: **for** $i \in [M]$ **do**
25:     **for** $j \in C_i$ **do**
26:       Learner sets $Best\_Arm[j] = B_i$
27:     **end for**
28: **end for**
29: **Return** $Best\_Arm$

**Algorithm 2** SE $(\mathcal{A}, \gamma, R)$

1: **Input:** $\mathcal{A}, \gamma, R$; **Initialize:** $\mathcal{A}_0 \leftarrow \mathcal{A}, r \leftarrow 0, \hat{\mu} \leftarrow 0_K$
2: **while** $|\mathcal{A}_r| > 1$ **and** $r < R$ **do**
3:     $r \leftarrow r+1, \epsilon_r = 2^{-r}$
4:     Pull each arm in $\mathcal{A}_{r-1}$ for $\frac{8\log(4|\mathcal{A}|r^2/\gamma)}{\epsilon_r^2}$ times
5:     Estimate $\hat{\mu}_k$ for all $k \in \mathcal{A}_{r-1}$ from these samples
6:     Set $\mathcal{A}_r \leftarrow \{i \in \mathcal{A}_{r-1} : \hat{\mu}_i \geq \max_{j \in \mathcal{A}_{r-1}} \hat{\mu}_j - \epsilon_r\}$
7: **end while**
8: **return** $\mathcal{A}_R, \hat{\mu}$

any two agents $i,j$ learning different bandits, $\exists\, k \in S_i \cup S_j$ s.t. $|\hat{\mu}_k^i - \hat{\mu}_k^j| > \eta/2$. Next, each agent $i$ communicates the quantities $S_i, \hat{\mu}^i$ to the learner (line 6), who uses this information to cluster the agents as described in lines 11-15. We will show that the above properties of the SE procedure ensure that with high probability, for each identified cluster, all its member agents are associated with the same bandit.

Lines $16 - 30$ describe the second phase of our scheme `Cl-BAI`. The learner selects one representative agent $a_i$ from every cluster $i$, and then instructs it to again call the successive elimination procedure SE, with input parameters $\mathcal{A} = S_i$, $\gamma = \delta/(2M)$, and $R = \infty$. This implies that for each representative agent, the SE procedure is run till there is only one arm left in the active set, which is then declared as the best arm estimate for the representative arm as well as all the other agents in the same cluster.

*Remark* 4.1 (Successive Elimination). We use successive elimination for its simplicity and *easy-to-tune* capability. We comment that in general other BAI (like track and stop (45)) and clustering ((20)) algorithms may also be used in this framework.

*Remark* 4.2 (Knowledge of separation $\eta$). We emphasize that the (exact) knowledge of separation $\eta$ may not be required. Any lower bound on $\eta$ is sufficient for theoretical results.

The following results demonstrate the correctness and sample complexity of `Cl-BAI`.

**Theorem 4.3.** *Suppose Assumption 2.1 holds. Given any* $\delta \in (0,1)$, *the* `Cl-BAI` *scheme (see Algorithm 1) is $\delta$-PC.*

**Theorem 4.4.** *With probability at least* $1 - \delta$, *the sample complexity of* `Cl-BAI` *for an instance* $\mathcal{I}$, *denoted by* $T_\delta^\mathcal{I}(Cl\text{-}BAI)$ *satisfies* $T_\delta^\mathcal{I}(CL\text{-}BAI) \leq T_1 + T_2$, *where*[4]

$$T_1 \lesssim \sum_{j \in [N]} \sum_{i=1}^{K} \max\{\Delta_{\mathcal{M}(j),i}, \eta\}^{-2} (\log K + \log N$$
$$+ \log\log\left(\max\{\Delta_{\mathcal{M}(j),i}, \eta\}^{-1}\right) + \log(1/\delta)),$$
$$T_2 \lesssim \sum_{j \in [M]} \sum_{i=1}^{K} \Delta_{j,i}^{-2}\left(\log K + \log M + \log\log\left(\Delta_{j,i}^{-1}\right) + \log(1/\delta)\right).$$

---
[4]We use $a \lesssim b$ to denote $a \leq Cb$, where $C$ is a positive constant.

each bandit problem $j \in [M]$, there is one cluster consisting of all the agents learning bandit $j$. In the second phase, the learner chooses one representative agent from each cluster, finds the best arm for that agent, and then declares that arm as the best arm for all the agents in the corresponding cluster.

Lines 3-15 describe the first phase where agents sample arms so that at the end of the phase, the learner can accurately map each agent to the bandit problem it is learning. To do this efficiently, each agent uses the successive elimination procedure $SE$ (43; 44) described in Algorithm 2.

For agent $i$, we will use $S_i$ and $\hat{\mu}^i = (\hat{\mu}_1^i, \hat{\mu}_2^i, ..., \hat{\mu}_K^i)$ to denote the set of surviving active arms and the vector of updated empirical mean rewards as returned by the SE procedure, respectively. We will prove in Appendix 10.2 that the SE procedure in the first phase, when run with suitable choices for $\gamma$ and $R$, guarantees the following with high probability: (i) For any two agents $i,j$ learning the same bandit, $\forall\, k \in S_i \cup S_j$, we have $|\hat{\mu}_k^i - \hat{\mu}_k^j| \leq \eta/2$; (ii) For

Note that $T_1$ and $T_2$ represent upper bounds on the total number of arm pulls in the first and second phases, respectively. In particular, recall that the second phase involves solving the standard best arm identification problem for $M$ bandits at one representative agent each and with target error probability of $\delta/(2M)$; this problem has been studied extensively and the expression directly follows from the literature, see for example (44).

*Remark* 4.5 (Comparison with a naive algorithm). We can compare the sample complexity of Cl-BAI with a naive single-phase scheme where the learner instructs each agent to independently identify their best arm using successive elimination, and then communicate the result back to the agent. Note that this scheme completely ignores the underlying mapping of agents to bandits. It follows immediately from (26) that for any agent learning bandit $j$, the sample complexity for the naive scheme is of the order of $\sum_{i=1}^{K} \Delta_{j,i}^{-2}$. Let us also assume balanced clusters, i.e., the cluster sizes are[5] $\Theta(N/M)$. Thus the overall average (normalized) sample complexity of the naive scheme across all agents is given by $NK \cdot \frac{1}{MK} \cdot \sum_{j\in[M]} \sum_{i=1}^{K} \Delta_{j,i}^{-2} := NK \cdot \bar{\Delta}^{-2}$, where $\bar{\Delta}$ can be thought of as representing the average problem complexity across the $M$ bandits.

On the other hand, from Theorem 4.4, the dominant terms in the (normalized) sample complexity of Cl-BAI are given by $NK \cdot \frac{1}{MK} \cdot \sum_{j\in[M]} \sum_{i=1}^{K} \max\{\Delta_{j,i}, \eta\}^{-2} + MK \cdot \bar{\Delta}^{-2}$. Comparing, we can see that the first term may be smaller than that of the naive algorithm since it involves terms of the form $\max\{\Delta_{j,i}, \eta\}^{-2}$, which is at most $\Delta_{j,i}^{-2}$ that appears in the naive scheme. In fact, it can be much smaller depending on the value of the 'separability' parameter $\eta$ from Assumption 2.1 vis-a-vis the bandit gaps; in particular when $\eta \gg \bar{\Delta}$. The second term in the above expression of the sample complexity of Cl-BAI will be much smaller whenever $M \ll N$, i.e., the number of bandits $M$ is much smaller than the number of agents $N$. We anticipate these conditions to be true in many scenarios of interest and thus expect our proposed scheme Cl-BAI to outperform the naive strategy. Our numerical experiments in Section 8 validate this intuition.

*Remark* 4.6 (Communication Cost). Next, we consider the communication complexity of Cl-BAI. Starting with communication from the agents to the central learner, it happens once in the first phase (line 6) where every agent communicates the active set and the empirical reward vector to the learner, thus resulting in a total cost of at most $O(N.(c_b.K + c_r.K))$ units; and then once in the second phase (line 23) where the selected representative from each cluster communicates the identity of its best arm to the learner, requiring a total cost of $O(c_b.M.\log K)$ units. Communication from the learner to agents happens only once in the second phase (line 18) when the learner selects a

representative agent from each cluster and instructs it to run Successive Elimination. This incurs a cost of $O(c_b.M)$ units.

Summing up and using $M \leq N$, $c_b \leq c_r$, the total communication cost required by Cl-BAI is at most $O(N.K.c_r)$ units. In comparison, the communication cost of a naive scheme, where each agent independently identifies their best arm and then communicates the result to the learner, is at most $O(N.\log K.c_b)$ units. Thus, Cl-BAI helps reduce the sample complexity by introducing interaction between the learner and the agents, which naturally induces a higher communication cost.

## 5. Algorithm II: **BAI-Cl**

Our second algorithm, Best Arm Identification then Clustering (BAI-Cl) is presented in Algorithm 3 and it also consists of two phases. In the first phase, the goal is to identify the set of best arms, i.e., $\{k_m^* : m \in [M]\}$. This is done by sampling agents randomly and finding their best arm, till we have identified $M$ different best arms. In the second phase, we aim to 'cluster' the remaining agents which were not sampled in the first phase and find the best arm corresponding to each of them. For each such agent, we do so by applying successive elimination only on the set of best arms identified in the first phase.

In the first phase (described in lines 3-17) the learner samples an agent $i$ uniformly at random from the set $A$, and then communicates the current set of best arms $S$ to agent $i$. The agent then proceeds to apply successive elimination (as prescribed in Algorithm 2) on the set of arms $[K]$ so that at the end of the SE procedure, we are confident that a) the returned set $S_i$ contains the best arm for agent $i$; and b) it does not contain the best arm corresponding to any of the other bandit instances.

The agent considers the intersection $S \cap S_i$. If it is non-empty, then it implies that another agent with the same best arm as agent $i$ had been sampled previously. In fact, the intersection will then have exactly one arm with high probability, corresponding to the best arm for agent $i$, and hence its index is communicated to the learner. On the other hand, if $S \cap S_i = \phi$ it means that the bandit that current agent is learning hasn't been explored yet. Hence, agent $i$ continues to run successive elimination on the set of arms $S_i$ (line 12) till only one arm remains, which is guaranteed to be the best arm for the agent with high probability and hence its index is communicated to the learner. The sets $A$ and $S$, as well as the array $Best\_Arm$ are updated appropriately.

At the end of the first phase, the set $S$ contains the indices of the $M$ best arms corresponding to the different bandit instances. What remains is to identify for each remaining agent in $A$, its corresponding best arm from within the set $S$.

Lines $18-25$ describe the second phase of our algorithm,

---

[5]We say $x = \Theta(y)$ if there exists positive constants $C_1$ and $C_2$ such that $C_1 y \leq x \leq C_2 y$.

**Algorithm 3** BAI-Cl

1: **Input:** $\eta,\delta$
2: **Initialize:** $A\leftarrow[N],S\leftarrow\phi,Best\_Arm\leftarrow0_N$
3: **First Phase:**
4: **while** $|S|<M$ **do**
5:     Learner samples agent $i$ from $A$ uniformly at random; communicates set $S$ to $i$
6:     Agent $i$ runs Successive Elimination:
7:     $S_i,\hat{\mu}^i=SE([K],\gamma=\frac{\delta.\log(\frac{M}{M-1})}{\log(\frac{3.M}{\delta})},R=\log(1/\eta)+1)$
8:     **if** $S\cap S_i\neq\phi$ **then**
9:         Agent $i$ sends arm $a_i^*\in S\cap S_i$ to learner
10:     **else**
11:         Agent $i$ further runs Successive Elimination:
12:         $a_i^*,\hat{\mu}^i=SE(S_i,\gamma=\frac{\delta.\log(\frac{M}{M-1})}{\log(\frac{3.M}{\delta})},R=\infty)$
13:         Agent $i$ sends arm $a_i^*$ to the learner
14:     **end if**
15:     Update at learner:
16:     $A=A\setminus\{i\},Best\_Arm[i]=a_i^*,S=S\cup a_i^*$
17: **end while**
18: **Second Phase:**
19: **for** $i\in A$ **do**
20:     Learner communicates the set $S$ with agent $i$
21:     $a_i^*,\hat{\mu}^i=SE(S,\delta/3N,R=\infty)$
22:     Agent $i$ communicates arm $a_i^*$ to the learner
23:     Learner sets $Best\_Arm[i]=a_i^*$
24: **end for**
25: **Return** $Best\_Arm$

where the learner communicates the set $S$ to the remaining agents in the set $A$. Each such agent applies successive elimination on the set $S$ with target error probability $\gamma=\delta/3N$, till the best arm is identified; and then communicates the arm index to the learner.

*Remark* 5.1 (Coupon Collector). The first phase ends when we see at least one agent from all $M$ bandits. This is related to the classical coupon collector problem, and we use those results to ensure that $\mathcal{O}(M\log M)$ agents will be sampled in this phase woth high probability.

The following result demonstrates the correctness of our proposed scheme BAI-Cl. The proof can be found in Appendix 10.

**Theorem 5.2.** *Given any $\delta\in(0,1)$, the BAI-Cl scheme (see Algorithm 3) is $\delta$-PC.*

**Theorem 5.3.** *Suppose each agent belongs to one of the $M$ clusters uniformly at random. Then, with probability at least $1-\delta$, the sample complexity $T_\delta^{\mathcal{I}}(BAI\text{-}Cl)$ satisfies*

$T_\delta^{\mathcal{I}}(BAI\text{-}Cl)\leq T_1+T_2$, *where,*

$$T_1\lesssim[\log K+\log\gamma+\log\log\Delta_{m,i}^{-1}]\{\sum_{m=1}^M\sum_{i=1}^K\Delta_{m,i}^{-2}$$
$$+M.\log(\frac{3.M}{\delta}).\max_{m\in[M]}\{\sum_{i=1}^K\max(\eta,\Delta_{m,i})^{-2}\}$$
$$T_2\lesssim N.M.\eta^{-2}(\log M+\log\delta^{-1}+\log N+\log\log\eta^{-1})$$

$T_1$ and $T_2$ denote the no. of pulls in first phase and second phase respectively. $T_1$ involves the pulls assigned for finding the best $M$ arms which is given by (ignoring log factors), $\sum_{m=1}^M\sum_{i=1}^K\Delta_{m,i}^{-2}$ along with that we will have at-most $M.\log(3M/\delta)$ agents, learning a bandit already been explored earlier. $T_2$ includes the pulls from applying SE on the set $S$ for at-most $N$ agents with $\delta/3N$ error probability.

*Remark* 5.4 (Comparison between BAI-Cl and Cl-BAI). We saw previously in Remark 4.5 that the dominant terms in the sample complexity of Cl-BAI are given by $\sum_{j\in[N]}\sum_{i=1}^K\max\{\Delta_{\mathcal{M}(j),i},\eta\}^{-2}+M.K.\bar{\Delta}^{-2}$. On the other hand, from Theorem 5.3, we have that the dominant terms in the sample complexity of BAI-Cl are given by $M.K.\bar{\Delta}^{-2}+M.\max_{m\in[M]}\{\sum_{i=1}^K\max(\eta,\Delta_{m,i})^{-2}\}+N.M.\eta^{-2}$. If the underlying instance is such that $\eta$ is large enough; in particular say that in each bandit there is at least a sizeable fraction of the arms whose mean reward is within $\eta$ of the corresponding best arm. Then we have, $\sum_{i=1}^K\max\{\Delta_{\mathcal{M}(j),i},\eta\}^{-2}\sim\Theta(K\eta^{-2})$, so that the respective complexities of Cl-BAI and BAI-Cl become $N.K.\eta^{-2}+M.K.\bar{\Delta}^{-2}$ and $N.M.\eta^{-2}+MK\eta^{-2}+M.K.\bar{\Delta}^{-2}$ respectively. Clearly, the main difference is between $N.K.\eta^{-2}$ for Cl-BAI and $(N+K).M.\eta^{-2}$ for BAI-Cl. Thus, BAI-Cl will perform much better whenever $M\ll N,K$, i.e., the number of bandits is much smaller than the number of agents and arms, which is a natural scenario. Another point to note is that while we expect BAI-Cl to perform better than Cl-BAI in most cases, the latter has the advantage that the agent pulls in the first phase happen in parallel which can sometimes be advantageous.

*Remark* 5.5 (Communication Cost). Communication from the learner to the agents happen in the first phase (line 5) where learner sequentially samples an agent and communicates the current set $S\subset[K]$ (of size at most $M$) to the agent; and then in the second phase where the learner communicates the final set $S$ (of size $M$) to all the remaining agents resulting a total cost of $O(c_b.N.\log(\sum_{i=0}^M\binom{K}{i}))=O(c_b.N.M.\log K)$ units. The communication from agents to the learner happens in the first phase (line 9 or 13) where each sampled agent incurs a cost of $c_b.\lceil\log K\rceil$. Since $O(M\log M)$ agents are sampled in the first phase with high probability, the total cost incurred is $O(c_b.M\log M.\log K)$. In the second phase (line 24), each remaining agent indicates one amongst the $M$ arms in $S$ as their best arm, requiring a total cost of

$O(c_b.N.\log M)$. Using $M \le N, K$, the total communication cost is at most $O(N.M.\log K.c_b)$ units. Comparing this with Cl-BAI which incurs a communication cost of $O(N.K.c_r)$ units (see Remark 4.6), we note that BAI-Cl is more communication-efficient (in addition to being better in terms of sample complexity) whenever $M \log K \le K$.

*Remark* 5.6 (Non-uniform Clusters). We assume that the each agent belongs to one of $M$ clusters uniformly. This can be easily generalized as this is equivalent to solving a coupon collector problem with unequal probabilities (see (46)).

# 6. Improved BAI-Cl: BAI-Cl++

Recall Assumption 2.1 that requires any admissible instance to satisfy a 'separability' constraint. In this section, we present a variant of BAI-Cl which requires an additional assumption other than Assumption 2.1, but can provide significant savings in terms of sample complexity.

**Assumption 6.1.** $\exists\ \eta_1 \ge 0$ such that for any two bandits $i$, $j$, the performance of the best arm $k_i^*$ of bandit $i$, differs by at least $\eta_1$ under bandit $j$, i.e, $|\mu_{i,k_i^*} - \mu_{j,k_i^*}| \ge \eta_1, \forall i,j \in [M], i \ne j$.

BAI-Cl++ is identical to BAI-Cl except that (i) in the first phase, after running $SE$ procedure(line 12) agent $i$ will pull arm $a_i^*$, $\frac{32\log(12M/\delta)}{\eta_1^2}$ times and communicates to the learner the estimated mean reward associated with $a_i^*$; and (ii) in the second phase (line 21 of Algorithm 3), it uses $\widehat{SE}$ (Algorithm 4) instead of the SE procedure. We will assume that at the end of the first phase, the learner stores the identified $M$ best arms and their estimated mean rewards in $S$ and $\overline{\mu}_S$ respectively. The $\widehat{SE}$ procedure uses Assumption 6.1 to provide a more efficient scheme for identifying the best arm amongst the set $S$ for each agent.

We defer the formal guarantees of BAI-Cl++ to Appendix 10.1. Similar to BAI-Cl, BAI-Cl++ is $\delta$-PC. Regarding sample complexity, in the first phase, both BAI-Cl++ and BAI-Cl require similar pulls (in fact BAI-Cl++ requires $M\eta_1^{-2}$ more pulls than BAI-Cl). However, in the second phase, instead of, $N.M.\Delta^{-2}(\log M + \log\delta^{-1} + \log N + \log\log\Delta^{-1})$ in BAI-Cl we have, $N.M.\Delta^{-2}(\log M + \log\log\Delta^{-1}) + N.\eta_1^{-2}(\log\delta^{-1} + \log N)$ . Hence, we will gain in no. of pulls using BAI-Cl++ over BAI-Cl as long as $N.M.\Delta^{-2} \ge (N+M)\eta_1^{-2}$.

# 7. Lower Bound

In this section, consider the class of problem instances $\mathcal{I}$ which in addition to Assumption 2.1, also satisfy the following condition on the mean reward gaps for each bandit $m \in [M]$: $\mu_{m,k_m^*} \ge \mu_{m,k} + \Delta, \forall k \ne k_m^*$. Also, we will restrict attention to unit variance Gaussian rewards for simplicity, although this can be readily generalized.

---

**Algorithm 4** $\widehat{SE}(S, \overline{\mu}_S, \gamma, \eta, \eta_1)$

1: **Input:** $S, \overline{\mu}_S, \gamma, \eta, \eta_1$; **Initialize:** $k \leftarrow 1$
2: **while** True **do**
3: $\quad \delta_k = 10^{-k}, \hat{a}, \hat{\mu} = SE(S, \delta_k, R = \log(1/\eta) + 1)$
4: $\quad$ Pull $\hat{a}$ for $32\frac{\log(4.k^2/\gamma)}{\eta_1^2}$ times
5: $\quad$ **if** $|\hat{\mu}_a - \overline{\mu}_S^a| < \eta_1/2$ **then**
6: $\quad\quad$ **return** $\hat{a}$ as the best arm
7: $\quad$ **else**
8: $\quad\quad k \leftarrow k+1$
9: $\quad$ **end if**
10: **end while**

---

We have the following lower bound on the expected sample complexity of any $\delta$-PC scheme over the class of instances $\mathcal{I}$.

**Theorem 7.1.** *For any $\delta$-PC algorithm $\mathcal{A}$, there exists a problem instance $\nu \in \mathcal{I}$ such that the expected sample complexity $\mathbb{E}[T_\delta^\nu(\mathcal{A})]$ satisfies*

$$\mathbb{E}[T_\delta^\nu(\mathcal{A})] \gtrsim \max\{M \cdot (K-M), N\}\frac{\log(1/\delta)}{\Delta^2}. \quad (1)$$

*Proof Sketch:* Let $\nu \in \mathcal{I}$ be an instance for which the mean reward vector corresponding to bandit $i$ satisfies, $\mu_{i,i} = \mu + \Delta, \mu_{i,j} = \mu \forall j \ne i$. Note that the best arm for bandit $i$ is arm $i$ under instance $\nu$. With a perturbed instance and *change of measure* (30, Lemma 1) argument we conclude the proof (details in Appendix 10.9).

*Remark* 7.2 (Orderwise Optimaity of BAI-Cl++). The above described class of Instance $\mathcal{I}$ satisfies Assumptions 2.1 and 6.1 with parameter $\Delta$ for both. For BAI-Cl++, the order-wise sample complexity is $M.K.\Delta^{-2} + (M+1)N\Delta^{-2}$. We compare it with Equation 1. Suppose $N \gg K, M$ and moreover $M$ is a constant (i.e., $M = \Theta(1)$). In that setup, the dominating term in the sample complexity of BAI-Cl++ is $N\Delta^{-2}$ which matches the lower bound (Equation 1). Hence BAI-Cl++ is order-wise optimal in this setting.

*Remark* 7.3 (Instance Dependent Lower Bound). The lower bound proposed in Theorem 7.1 is a worst-case bound. Using similar ideas, we can also derive an instance-dependent lower bound which is more general but requires additional notation. Details can be found in Appendix 10.10.

# 8. Numerical Results

We conduct an empirical evaluation of our proposed algorithms using both synthetic and real-world datasets. We set the error probability $\delta = 10^{-10}$ for our experiments and present sample complexity results which are averaged over multiple independent runs of the corresponding algorithms.

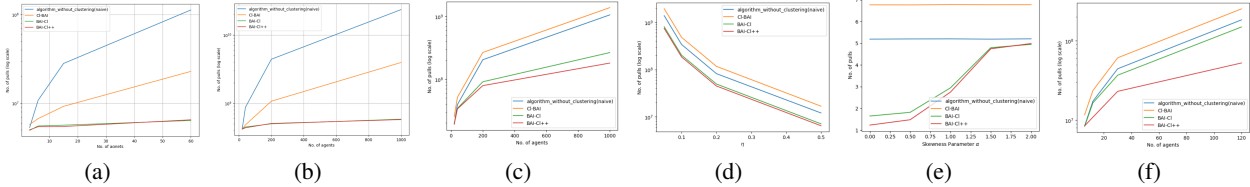

*Figure 1.* (a)(b)(c) Performance with varying number of agents $N$ for synthetic datasets 1, 2, 3 (d) Varying clustering parameter $\eta$ for dataset 3 (e) Skewed cluster sizes for dataset 3 (f) Performance with varying number of agents $N$ for MovieLens dataset

### 8.1. Synthetic Datasets

We take the reward distribution for each arm to be unit-variance Gaussian. We consider three problem instances.

**First:** We consider a small instance with $M = 3$ bandits/clusters and $K = 10$ arms, with mean arm rewards for the three bandits given by $\boldsymbol{\mu_1} = [.09, .26, .49, .91, .56, .16, .31, .75, .76, .77]$; $\boldsymbol{\mu_2} = [.02, .27, .36, .42, .47, .92, .32, .62, .82, .9]$; and $\boldsymbol{\mu_3} = [.14, .46, .64, .44, .7, .03, .96, .72, .79, .95]$. The best arms for the bandits are arms 4, 6, and 7 respectively, and both Assumptions 2.1 and 6.1 hold with parameter $\eta = \eta_1 = 0.3$.

**Second:** Next, we consider a larger instance with $M = 20$ bandits/clusters and $K = 100$ arms. There is a unique best arm for each of the $M$ bandit problems. For each bandit, the mean reward for the assigned best arm is sampled from a uniform distribution $U(1-\eta, 1)$. Next, the means of the $M-1$ arms that are best for other bandits are sampled from $U(0, 1-2\eta)$. For the remaining $K - M$ arms, their mean rewards are sampled uniformly between 0 and the mean reward of the best arm. It can be verified that Assumptions 2.1 and 6.1 are satisfied with parameter $\eta$. We set $\eta = 0.15$.

**Third:** We again set $M = 20$, $K = 100$, with a unique best arm for each bandit having mean reward 1, while all other arms have mean $1-\eta$. Again, both Assumptions 2.1 and 6.1 hold with parameter $\eta$. We set $\eta = 0.15$.

Finally, there are $N$ agents, divided into $N/M$ sized clusters.

**Variation with number of agents ($N$):** For the three datasets constructed above, we vary $N$ and plot the average number of pulls for the various schemes in Figures 1(a)(b)(c). We observe that `BAI-Cl` and `BAI-Cl++` perform the best for all the datasets. For datasets 1 and 2, `Cl-BAI` also provides a significant improvement over the naive single-phase scheme (with no clustering). This is in line with Remarks 4.5 and 5.4 which suggest that `Cl-BAI` performs better than the naive scheme when the clustering parameter $\eta$ is large as compared to the individual bandit arm reward gaps. For example, for dataset 1, $\eta$ is .3, while the minimum arm mean reward gaps for the three bandits are .14, .02, .01 respectively. On the other hand, for dataset 3 both the naive scheme and `Cl-BAI` have poor performance. Again, this is consistent

with Remark 4.5 since both the clustering parameter and the individual bandit arm reward gaps are $\eta$ in this case.

**Variation in clustering parameter ($\eta$):** For dataset 3, we varied $\eta$ over the set $[0.05, 0.1, 0.2, 0.5]$, while keeping other parameters constant as follows: $K = 100$, $M = 20$, and $N = 100$. From Figure 1(d), we see that the sample complexity decays rapidly as the clustering parameter $\eta$ increases and thus the underlying problem instance becomes easier.

**Variation in cluster sizes:** While the previous experiments assume that all clusters are of the same size, here we study the impact of non-uniformity of cluster sizes on the performance of the various algorithms. We consider dataset 3 with $N = 500$ agents and $M = 100$ clusters, where the cluster sizes follow a power-law distribution. In particular, each agent is mapped to cluster $i$ with probability proportional to $i^\alpha$, where $\alpha$ governs the skewness of the cluster sizes. As $\alpha$ increases, the cluster sizes become more skewed. Figure 1(e) presents the average number of pulls for the various schemes as $\alpha$ is varied. While the sample complexity of the naive scheme and `Cl-BAI` is invariant to $\alpha$, the performance of `BAI-Cl` and `BAI-Cl++` worsens as $\alpha$ increases. This is because these algorithms are required to identify all the best arms in the first phase by randomly sampling agents; and this task becomes significantly harder when there are clusters with much fewer agents as compared to others.

### 8.2. MovieLens Dataset

We perform experiments using the *MovieLens-1M* dataset, which contains movie ratings from a large number of users. We group the users into six age categories: 18–24, 25–34, 35–44, 45–49, 50–55, and 56+. The 0–18 age group is excluded due to insufficient ratings for many movies. We restrict our study to movies that received at least 30 ratings in each of the six age groups, leaving 316 movies.

We have $M = 6$ bandits, one corresponding to each age group. Each of the $K = 316$ movies represents an arm. For each bandit and arm pair, the reward distribution is taken to be the empirical average score distribution calculated from the reviews for the corresponding movie given by users in that age group, suitably normalized to make it 1-subGaussian.

We find that each of the 6 bandits (user age groups) has

a distinct best arm (movie with highest average rating). For example, the highest rated movie for the 18–24 group is "The Usual Suspects (1995)", while it is "To Kill a Mockingbird (1962)" for the 56+ age group. The dataset satisfies Assumptions 2.1 and 6.1 with clustering parameters $\eta = 0.0027$ and $\eta_1 = 0.026$ respectively.

As before, we assume that there are $N$ agents divided into $M$ equal-sized clusters. The goal of the learner is to identify the best arm (movie with highest expected score) for each agent.

Figure 1(f) plots the average sample complexity for the various schemes as we vary $N$. Our results demonstrate that clustering-based methods, especially `BAI-Cl++`, significantly reduce the sample complexity compared to the naive scheme. `BAI-Cl` also achieves competitive performance but is less efficient than `BAI-Cl++`.

### 8.3. Yelp Dataset

We conducted a similar experiment using the *Yelp* dataset as well. Those results along with some additional numerics can be found in Appendix 10.11.

## 9. Conclusions and Future Work

In this paper, we propose and analyze algorithms for best arm identification in a multi-agent multi-armed bandit setting, with agents grouped into (a priori unknown) clusters. Out of these, the algorithm `BAI-Cl` (actually, its improved version `BAI-Cl++`) obtains near optimal sample complexity, which is validated by lower bounds. One drawback of `BAI-Cl` is that it requires the knowledge of the cluster separation $\eta$. One immediate future work is to propose and analyze algorithms which do not require knowledge of $\eta$.

Towards this, we ran additional experiments with the Movie-lens and Yelp datasets, and observed that in practice, `BAI-Cl` is indeed robust to $\eta$. Theoretically we believe a successive estimation based algorithm may work, and here we sketch the rough idea. We can propose a multi-phase algorithm where we start with a large enough value of $\eta$, and at the beginning of each phase, we reduce $\eta$ by a factor of 2. So, for the first few phases, the algorithm will not perform well, since $\eta$ is still large enough. However, after some phases, the value of $\eta$ falls below the actual gap and the algorithms start learning the best arm. If we select exponentially increasing phase lengths, we can show that these multi-phase algorithm will succeed in finding the best arms of all the agents. Of course, the sample complexity of this algorithm will be worse compared to the current ones; however, we believe the orderwise sample complexity will remain the same (with worse constants) if the initial phase-length and the exponential increase factor are properly chosen.

However, we need to come-up with a stopping criteria for halving $\eta$, which may be crucial. Currently, we do not know how to do this appropriately and hence this is deferred as a future work.

Also, in this work we use Successive elimination for *easy* tunability. There are other best arm identification algorithms, such as Track And Stop (45), which are asymptotically optimal. An interesting future direction is to use such algorithms in the clustered multi-agent bandits framework.

## Acknowledgements

AG would like to acknowledge Amazon grant (RD/0123-AMAZO09-005). NK's work was supported by a SERB MATRICS grant and a CSR grant from SBI. Yash acknowledges funding support from the Centre for Machine Intelligence and Data Science (C-MInDS), IIT Bombay.

## Impact Statement

This paper presents work whose goal is to advance the field of Machine Learning, especially in the field for Federated and (distributed) multi-agent bandits. There are many potential societal consequences of our work, none which we feel must be specifically highlighted here.

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

# 10. Appendix

## 10.1. Guarantees for `BAI-Cl++`

We have the following results.

**Theorem 10.1.** *Given any $\delta \in (0,1)$, the BAI-Cl++ scheme is $\delta$-PC.*

**Theorem 10.2.** *Suppose each agent belongs to one of the $M$ clusters uniformly at random. With probability at least $1 - \delta$, the sample complexity $T_\delta^{\mathcal{I}}(BAI\text{-}Cl{++})$ of BAI-Cl++ for an instance $\mathcal{I}$ satisfies*

$$T_\delta^{\mathcal{I}}(BAI\text{-}Cl{++}) \leq T_1 + T_2, \text{ where } \gamma = \delta. \frac{\log(\frac{M}{M-1})}{\log(\frac{3.M}{\delta})} \text{ and}$$

$$T_1 \lesssim \sum_{m=1}^{M} \sum_{i=1}^{K} \Delta_{m,i}^{-2} (\log K + \log \gamma + \log\log \Delta_{m,i}^{-1})$$

$$+ M.\log(\frac{3.M}{\delta}). \max_{m \in [M]} \Big\{ \sum_{i=1}^{K} \max(\eta, \Delta_{m,i})^{-2} (\log K + \log \gamma$$

$$+ \log\log\max(\eta, \Delta_{m,i})^{-1}) \Big\} + M \left( \frac{\log(\delta^{-1}) + \log(M)}{\eta_1^2} \right)$$

$$T_2 \lesssim N.M.\eta^{-2} (\log M + \log\log \eta^{-1})$$

$$+ N.\eta_1^{-2} (\log \delta^{-1} + \log N)$$

*Remark* 10.3. Comparing the sample complexity of BAI-Cl++ to BAI-Cl. In the first phase we have an additional $M \left( \frac{\log(\delta^{-1}) + \log(M)}{\eta_1^2} \right)$ pulls since we want estimated mean of the best arm to be within $\eta_1$ of the true mean to get correct result from $\widehat{SE}$ procedure. In the second phase instead of,

$$N.M.\eta^{-2} (\log M + \log \delta^{-1} + \log N + \log\log \eta^{-1})$$

we have,

$$N.M.\eta^{-2} (\log M + \log\log \eta^{-1}) + N.\eta_1^{-2} (\log \delta^{-1} + \log N)$$

Hence, we will gain in no. of pulls using BAI-Cl++ over BAI-Cl as long as $N.M.\eta^{-2} \geq (N + M)\eta_1^{-2}$

*Remark* 10.4 (Communication Cost). Note that the modifications made to BAI-Cl to get BAI-Cl++ introduce additional communication in the first phase of the scheme. Each sampled agent, in addition to the identity of its identified best arm, also communicates an estimate of its mean reward to the learner. This incurs a total additional cost of at most $O(M.c_r)$ units. Thus, the overall communication cost for BAI-Cl++ is at most $O(N.M.\log K.c_b + M.c_r)$ units.

## 10.2. Proof of Successive Elimination

Assume there are $n$ arms $\{1, 2, \cdots, n\}$ in $\mathcal{A}$ with corresponding means $\mu_1 \geq \mu_2 \geq \cdots \geq \mu_n$, and are 1 sub-Gaussian. Also, let $\Delta_i = \mu_1 - \mu_i$ for $i \in \{2, 3, \cdots, n\}$ and $\Delta_1 = \Delta_2$. Consider the successive elimination procedure $SE$ applied to the set $\mathcal{A}$. The following result is well known (43; 44) and we include a proof here for completeness.

**Theorem 10.5.** *With probability at least $1 - \gamma$, $SE(\mathcal{A}, \gamma, R = \infty)$ satisfies the following properties:*

- *It returns the best arm in $\mathcal{A}$, i.e., arm $1$.*

- *The total number of arm pulls needed is at most*

$$O \left( \sum_{i=1}^{n} \frac{\log\left( \frac{n\log \Delta_i^{-1}}{\gamma} \right)}{\Delta_i^2} \right)$$

*Proof.* After $r$ rounds of Successive Elimination, the total no. of pulls for any surviving arm in the active set $\mathcal{A}_r$ is at least $8.\log(4nr^2/\gamma)/\epsilon_r^2$. Then using Hoeffding's inequality for sub-gaussian , we have the following bound on the difference

between the true mean $\mu_i$ of any arm $i$ and its empirical estimate $\hat{\mu}_i$ at the end of any round $r$.

$$\mathcal{P}\left(|\mu_i - \hat{\mu}_i| \geq \frac{\epsilon_r}{2}\right) \leq \frac{\gamma}{2nr^2}. \tag{2}$$

For each $r \geq 0$, define the event $E_r = \{1 \in \mathcal{A}_r$ and $j \notin \mathcal{A}_r \; \forall \, j \in \mathcal{A}$ s.t. $\mu_j < \mu_1 - 2\epsilon_r\}$. Clearly event $E_0$ holds true since the active set $\mathcal{A}_0$ is initialized to include all arms in the set $\mathcal{A}$. Furthermore, note that the event $E = \cap_{r=1}^{\infty} E_r$ refers to the event that only the best arm, i.e. arm 1, remains in the active set and is thus returned by the SE procedure.

We have

$$\mathcal{P}[E_r | \cap_{k=1}^{r-1} E_k] \geq \mathcal{P}[|\mu_i - \hat{\mu}_i| \leq \frac{\epsilon_r}{2} \; \forall i \in \mathcal{A}_{r-1} | \cap_{k=1}^{r-1} E_k]$$

$$\geq 1 - n. \frac{\gamma}{2nr^2} = 1 - \frac{\gamma}{2r^2}$$

where the last inequality follows from (2). Next,

$$\mathcal{P}(E) = \prod_{r=1}^{\infty} \mathcal{P}[E_r | \cap_{k=1}^{r-1} E_k]$$

$$\geq \prod_{r=1}^{\infty} \left(1 - \frac{\gamma}{2r^2}\right)$$

$$\geq 1 - \sum_{r=1}^{\infty} \frac{\gamma}{2r^2}$$

$$\geq 1 - \gamma$$

Thus we have that with probability at least $1-\gamma$, $SE(\mathcal{A}, \gamma, R=\infty)$ returns the best arm in $\mathcal{A}$, i.e., arm 1. Next, we will now prove the upper bound on the number of pulls required by the SE procedure. Note that with probability at least $1-\gamma$, each arm $i \geq 2$ is removed from the active set by at most round $\lceil 1 + \log_2 \frac{1}{\Delta_i} \rceil$. Also, the number of pulls of arm 1 is at most the number of pulls of any other arm. Thus, the total number of pulls for the SE procedure is at most

$$\sum_{i=1}^{n} \sum_{r=1}^{\lceil \log_2 \frac{1}{\Delta_i} \rceil} \frac{8.\log(\frac{4nr^2}{\gamma})}{2^{-2r}}$$

$$\leq O\left(\sum_{i=1}^{n} \frac{\log\left(\frac{n\log\Delta_i^{-1}}{\gamma}\right)}{\Delta_i^2}\right). \tag{3}$$

$\square$

## 10.3. Proof of Theorem 1

**Proposition 10.6.** *Let $\hat{\mu}_{k,r}^i$ denote the estimated mean of $k^{th}$ arm of $i^{th}$ agent at $r^{th}$ round. Consider the bad event $e_0$, which occurs if in the first phase for any round $t$ of Successive Elimination for any agent $i$ and for any arm $k$, $|(\mu_{\mathcal{M}(i),k}) - (\hat{\mu}_{k,r}^i)| \geq \epsilon_r/2$ happens. Using union bound and theorem 10.5 we have*

$$P(e_0) \leq N.\gamma$$

**We will assume that $e_0$ does not occur with probability $1 - N\gamma$ and proceed with our proof.**

**Proposition 10.7.** *Consider the bad event $e_1$ to be when there exist two agents learning the same bandit and are clustered in different clusters. Then*

$$P(e_1 \cap \bar{e}_0) \leq N.K\left(2\sqrt{2}. \frac{\gamma^{0.75}}{K^{0.75}} + \frac{\gamma}{K}\right) \tag{4}$$

*Proof.* Two agents, $i$ and $j$, learning the same bandit will not be assigned to the same cluster if $D(\hat{\mu}^i, \hat{\mu}^j) \geq \frac{\eta}{2}$, i.e., there exists an arm in the union of the active sets of these two agents, whose estimated means for agents $i$ and $j$ differ by more than $\frac{\eta}{2}$.

**Proof sketch:** We want to prove that the probability that there exist agents $i$ and $j$, and an arm $k$ in $S_i \cup S_j$ such that $|\hat{\mu}_k^j - \hat{\mu}_k^i| \geq \eta/2$, is less than $NK\left(2\sqrt{2} \cdot \frac{\gamma^{0.75}}{K^{0.75}} + \frac{\gamma}{K}\right)$. We first prove the following claims.

Claim 1 states that, given $\bar{e}_0$, for any agent $i$, all the arms in $S_i$ will be "good" arms. We define the set of "good" arms for agent $i$, denoted as $G_i$, as the set of arms whose true mean is within $2\epsilon_R$ of the mean of the best arm for agent $i$.

From Claim 2, we conclude that $e_1 \cap \bar{e}_0$ implies there must exist an agent $i$ and an arm $k \in G_i$ such that the estimate of the $k^{th}$ arm for the $i^{th}$ agent is at least $\eta/2 - \epsilon_R/2$ less than its true mean.

Claim 3 puts an upper bound on the probability that a "good" arm gets eliminated after $r$ rounds of successive elimination.

Finally, we combine Claims 1, 2, and 3 to obtain the upper bound on the event $e_1$.

*Claim* 1. For all $i \in [N]$ and $k \in S_i$, we have $\mu_{\mathcal{M}(i),k} \geq \mu_{\mathcal{M}(i),k^*_{\mathcal{M}(i)}} - 2\epsilon_R$.

*Proof.* All arms with a true mean at least $2\epsilon_R$ less than the mean of the best arm will be eliminated after round $R$, i.e., all the arms in the set

$$E = \{k \,|\, \mu_{\mathcal{M}(i),k} \leq \mu_{\mathcal{M}(i),k^*_{\mathcal{M}(i)}} - 2\epsilon_R\} \tag{5}$$

From Proposition 10.6, for $k \in E$, we have:

$$|\mu_{\mathcal{M}(i),k} - \hat{\mu}_{k,R}^i| \leq \epsilon_R/2,$$
$$|\hat{\mu}_{k^*_{\mathcal{M}(i)},R}^i - \mu_{\mathcal{M}(i),k^*_{\mathcal{M}(i)}}| \leq \epsilon_R/2,$$
$$\text{Hence,} \quad |\hat{\mu}_{k^*_{\mathcal{M}(i)},R}^i - \hat{\mu}_{k,R}^i| \geq 2\epsilon_R - \epsilon_R/2 - \epsilon_R/2 \geq \epsilon_R.$$

$\square$

*Claim* 2. $e_1 \cap \bar{e}_0 \longrightarrow$ There exists a pair of agents $i,j$ with $\mathcal{M}(i) = \mathcal{M}(j)$ and an arm $k \in S_i \cup S_j$, such that

$$\hat{\mu}_k^i \leq \mu_{\mathcal{M}(i),k} - \left(\frac{\eta}{2} - \frac{\epsilon_R}{2}\right),$$
$$\text{or}$$
$$\hat{\mu}_k^j \leq \mu_{\mathcal{M}(j),k} - \left(\frac{\eta}{2} - \frac{\epsilon_R}{2}\right).$$

*Proof.* $e_1$ implies that there exists a pair of agents $i,j$ such that $\mathcal{M}(i) = \mathcal{M}(j)$ and some arm $k \in S_i \cup S_j$ for which the difference in estimated means of that arm between the two agents $i,j$ is more than $\eta/2$, i.e.,

$$|\hat{\mu}_k^i - \hat{\mu}_k^j| \geq \eta/2.$$

Since $k$ must belong to either $S_i$ or $S_j$, let's assume $k \in S_j$. Then, from Claim 1, we have $k \in G_i = G_j$. Also, from Proposition 10.6, we have

$$|\mu_{\mathcal{M}(j),k} - \hat{\mu}_k^j| \leq \epsilon_R/2,$$

and hence

$$|\mu_{\mathcal{M}(i),k} - \hat{\mu}_k^i| \geq \frac{\eta}{2} - \frac{\epsilon_R}{2}.$$

This implies:

$$\Rightarrow \hat{\mu}_k^i - \mu_{\mathcal{M}(i),k} \leq -\left(\frac{\eta}{2} - \frac{\epsilon_R}{2}\right), \tag{6}$$
$$\text{or}$$
$$\Rightarrow \hat{\mu}_k^i - \mu_{\mathcal{M}(i),k} \geq \left(\frac{\eta}{2} - \frac{\epsilon_R}{2}\right). \tag{7}$$

But Equation (7) isn't possible under the good event $\bar{e}_0$, because arm $k$ must get eliminated from agent $i$ in some round $r \in [1,R]$. If arm $k$ is in both $S_i$ and $S_j$, then

$$|\hat{\mu}_k^i - \hat{\mu}_k^j| \leq \epsilon_R,$$

and hence, an error cannot occur due to this arm. Arm $k$ gets eliminated in round $r$ only if there exists some arm $k'$ whose estimated mean is greater than $\hat{\mu}_k^i + \epsilon_r$ (see line 7 of Algorithm 2). This implies:

$$\hat{\mu}_{k'}^i \geq \hat{\mu}_k^i + \epsilon_r,$$

$$\mu_{\mathcal{M}(i),k'} + \frac{\epsilon_r}{2} \geq \mu_{\mathcal{M}(i),k} + \frac{\eta}{2} - \frac{\epsilon_R}{2} + \epsilon_r,$$

$$\geq \mu_{\mathcal{M}(i),k} + \frac{17\epsilon_R}{2} - \frac{\epsilon_R}{2} + \frac{\epsilon_r}{2},$$

$$> \mu_{\mathcal{M}(i),k} + 2\epsilon_R. \tag{8}$$

Equation (8) violates Claim 1, hence (7) isn't possible. Therefore, Equation (6) proves our claim.

$\square$

*Claim* 3. For any arm $k$ of agent $i$ satisfying $\mu_{\mathcal{M}(i),k} > \mu_{\mathcal{M}(i),k^*_{\mathcal{M}(i)}} - 2\epsilon_R$, the probability that this arm gets eliminated after $r$ rounds, assuming $\bar{e}_0$ holds, is less than

$$e^{-4\frac{\log\left(\frac{3Kr^2}{\gamma}\right)\left(\frac{\epsilon_r}{2} - 2\epsilon_R\right)^2}{\epsilon_r^2}}.$$

*Proof.* Arm $k$ can be removed from agent $i$ after $r$ rounds only if

$$\hat{\mu}_{k,r}^i < \hat{\mu}_{k',r}^i - \epsilon_r \quad \text{for some arm } k'.$$

From Proposition 10.6, we know that

$$|\hat{\mu}_{k',r}^i - \mu_{\mathcal{M}(i),k'}| \leq \frac{\epsilon_r}{2}.$$

Therefore, we can deduce:

$$\hat{\mu}_{k,r}^i \leq \mu_{\mathcal{M}(i),k} - \frac{\epsilon_r}{2} + 2\epsilon_R. \tag{9}$$

After $r$ rounds, the total number of pulls for any arm is greater than $8 \cdot \log\left(\frac{4Kr^2}{\gamma}\right)/\epsilon_r^2$. Using Hoeffding's inequality, we can bound the probability of the event in equation (9) by:

$$\leq e^{-4\frac{\log\left(\frac{4Kr^2}{\gamma}\right)\left(\frac{\epsilon_r}{2} - 2\epsilon_R\right)^2}{\epsilon_r^2}}.$$

This concludes the proof for Claim 3. $\square$

From Claim 3, we have the probability of an arm getting eliminated in round $r$. Also, after round $r$, the total number of pulls of any arm is at least $8 \cdot \log\left(\frac{3Kr^2}{\gamma}\right)/\epsilon_r^2$. Hence, from Hoeffding's inequality, we can derive the following:

$$\mathcal{P}\left(\hat{\mu}_{k,r}^i \leq \mu_{\mathcal{M}(i),k} - \left(\frac{\eta}{2} - \frac{\epsilon_R}{2}\right)\right) \leq \exp\left(-4\log\left(\frac{4Kr^2}{\gamma}\right)\frac{\left(\frac{\eta}{2} - \frac{\epsilon_R}{2}\right)^2}{\epsilon_r^2}\right)$$

To bound the probability of $e_1 \cap \bar{e}_0$, using Claim 2, we can say that it is upper-bounded by the probability that **there exists an arm $k$ of agent $i$ and an agent $j$ satisfying $k \in S_i \cup S_j$ such that $\hat{\mu}_{k,r}^i \leq \mu_{\mathcal{M}(i),k} - \left(\frac{\eta}{2} - \frac{\epsilon_R}{2}\right)$**. We assume that arm $k$ gets eliminated after some round $r \in [1,R]$, since if it doesn't get eliminated after round $R$, its estimated mean will be within $\frac{\epsilon_R}{2}$, and thus $\hat{\mu}_{k,r}^i \leq \mu_{\mathcal{M}(i),k} - \left(\frac{\eta}{2} - \frac{\epsilon_R}{2}\right)$ won't hold, as $\epsilon_R = 2^{-\log(17/\eta)} = \eta/17$.

The number of pulls for an arm that gets eliminated after round $r$ will be at least $8 \cdot \log\left(\frac{4Kr^2}{\gamma}\right)/\epsilon_r^2$. Hence, the probability that after $r$ rounds arm $k$ gets eliminated and has an estimated mean $\frac{\eta}{2} - \frac{\epsilon_R}{2}$ less than its true mean is bounded by

$$\min\left(e^{-4\log\left(\frac{4Kr^2}{\gamma}\right)\frac{\left(\frac{\epsilon_r}{2} - 2\epsilon_R\right)^2}{\epsilon_r^2}}, e^{-4\log\left(\frac{4Kr^2}{\gamma}\right)\frac{\left(\frac{\eta}{2} - \frac{\epsilon_R}{2}\right)^2}{\epsilon_r^2}}\right)$$

Taking the union bound over all such arms $k$ and corresponding agent $i$, we obtain the following equation:

$$P(e_1 \cap \bar{e}_0) \leq N \cdot K \sum_{r=1}^{R} \min \left( e^{-4\log\left(\frac{4Kr^2}{\gamma}\right)\frac{\left(\frac{\epsilon_r}{2}-2\epsilon_R\right)^2}{\epsilon_r^2}}, e^{-4\log\left(\frac{4Kr^2}{\gamma}\right)\frac{\left(\frac{\eta}{2}-\frac{\epsilon_R}{2}\right)^2}{\epsilon_r^2}} \right)$$

$$\leq N \cdot K \sum_{r=1}^{R} e^{\min\left(-4\log\left(\frac{4Kr^2}{\gamma}\right)\frac{\left(\frac{\epsilon_r}{2}-2\epsilon_R\right)^2}{\epsilon_r^2}, -4\log\left(\frac{4Kr^2}{\gamma}\right)\frac{\left(\frac{\eta}{2}-\frac{\epsilon_R}{2}\right)^2}{\epsilon_r^2}\right)}$$

$$\leq N \cdot K \sum_{r=1}^{R} \left(\frac{\gamma}{4Kr^2}\right)^{\max\left(1-8\frac{\epsilon_R}{\epsilon_r}, 4\frac{\left(\frac{\eta}{2}-\frac{\epsilon_R}{2}\right)^2}{\epsilon_r^2}\right)}.$$

$\epsilon_R = 2^{-\log(17/\eta)} = \eta/17,$

$$1 - 8\frac{\epsilon_R}{\epsilon_r} \geq 0.75 \text{ for } r \leq R-5$$

$$4\frac{\left(\frac{\eta}{2}-\frac{\epsilon_R}{2}\right)^2}{\epsilon_r^2} \geq 1 \text{ for } r \geq R-4$$

Hence,

$$\leq N.K\left(\sum_{r=1}^{R-5}\left(\frac{\gamma}{4Kr^2}\right)^{0.75} + \sum_{r=R-4}^{R}\left(\frac{\gamma}{4Kr^2}\right)\right)$$

$$\leq N.K\left(2\sqrt{2}.\frac{\gamma^{0.75}}{K^{0.75}} + \frac{\gamma}{K}\right)$$

$\square$

**Proposition 10.8.** *Consider the event $e_2$ as the event where two agents learning different bandits get clustered into the same cluster, i.e., if $D(\hat{\mu}_i, \hat{\mu}_j) \leq \frac{\eta}{2}$, then*

$$P(e_2 \cap \bar{e}_0) \leq (M-1)N\left(\frac{\gamma}{K}\right). \tag{10}$$

*Proof.* **Proof sketch:** We want to bound the probability that any two agents who are learning different bandits get clustered in the same cluster. Throughout this proof, we consider the possibilities only under the event $\bar{e}_0$, and the probabilities are bounded as intersections with $\bar{e}_0$.

We first claim (Claim 4) that if there exist two agents $i, j$ who get clustered into the same cluster, then the estimated mean of arm $k^*_{\mathcal{M}(j)}$ for the $i$th user must satisfy

$$\hat{\mu}^i_{k^*_{\mathcal{M}(j)}} \geq \mu_{\mathcal{M}(j), k^*_{\mathcal{M}(j)}} - \frac{\eta}{2} - \frac{\epsilon_R}{2}.$$

Next, in Claim 5, we state that if $k^*_{\mathcal{M}(j)}$ from Claim 4 gets eliminated for user $i$ after $r$ rounds of successive elimination, then there must exist an arm $k' \in [K] \setminus \{k^*_{\mathcal{M}(j)}\}$ such that

$$\hat{\mu}^i_{k'} \geq \hat{\mu}^i_{k^*_{\mathcal{M}(j)}} + \epsilon_r.$$

We bound the probabilities of the events from Claims 4 and 5 using Hoeffding's inequality for sub-Gaussian random variables. Since the events from Claims 4 and 5 are independent, we can take the product of their probabilities to obtain an upper bound on the probability of their intersection.

$\square$

**Define** $e_2^{i,j}$ *as the event that agents $i$ and $j$, with different best arms $k^*_{\mathcal{M}(i)}$ and $k^*_{\mathcal{M}(j)}$, where $\mu_{\mathcal{M}(j),k^*_{\mathcal{M}(j)}} \geq \mu_{\mathcal{M}(i),k^*_{\mathcal{M}(i)}}$, are clustered into the same cluster.*

**Claim** 4. The event $e_2$ implies that there exists an agent $i$ and an arm $k^*_{\mathcal{M}(j)} \in \{k_1^*, k_2^*, \dots, k_M^*\} \setminus k^*_{\mathcal{M}(i)}$ where $\mu_{\mathcal{M}(j),k^*_{\mathcal{M}(j)}} \geq \mu_{\mathcal{M}(i),k^*_{\mathcal{M}(i)}}$, such that

$$\hat{\mu}^i_{k^*_{\mathcal{M}(j)}} \geq \mu_{\mathcal{M}(j),k^*_{\mathcal{M}(j)}} - \frac{\eta}{2} - \frac{\epsilon_R}{2}.$$

*Proof.* The event $e_2$ implies that there exist two agents $i,j$ such that $e_2^{i,j}$ holds. From the definition of $e_2^{i,j}$ and our condition for clustering we have,

$$e_2^{i,j} \implies \left( |\hat{\mu}^i_{k^*_{\mathcal{M}(i)}} - \hat{\mu}^j_{k^*_{\mathcal{M}(i)}}| < \frac{\eta}{2} \right) \wedge \left( |\hat{\mu}^j_{k^*_{\mathcal{M}(j)}} - \hat{\mu}^i_{k^*_{\mathcal{M}(j)}}| < \frac{\eta}{2} \right) \wedge \left( \mu_{\mathcal{M}(j),k^*_{\mathcal{M}(j)}} \geq \mu_{\mathcal{M}(i),k^*_{\mathcal{M}(i)}} \right). \tag{11}$$

From Assumption 2.1, we have

$$(\mu_{i,k^*_{\mathcal{M}(i)}} - \mu_{i,k^*_{\mathcal{M}(j)}}) \geq \eta.$$

Also, since $\mu_{j,k^*_{\mathcal{M}(j)}} \geq \mu_{i,k^*_{\mathcal{M}(i)}}$, it follows that

$$(\mu_{j,k^*_{\mathcal{M}(j)}} - \mu_{i,k^*_{\mathcal{M}(j)}}) \geq \eta.$$

From the equation above, we have

$$|\hat{\mu}^j_{k^*_{\mathcal{M}(j)}} - \hat{\mu}^i_{k^*_{\mathcal{M}(j)}}| < \frac{\eta}{2}.$$

Thus,

$$\hat{\mu}^i_{k^*_{\mathcal{M}(j)}} - \mu_{i,k^*_{\mathcal{M}(j)}} \geq (\mu_{j,k^*_{\mathcal{M}(j)}} - \mu_{i,k^*_{\mathcal{M}(j)}}) - \frac{\eta}{2} - \frac{\epsilon_R}{2} \quad \text{or} \quad |\hat{\mu}^j_{k^*_{\mathcal{M}(j)}} - \mu_{j,k^*_{\mathcal{M}(j)}}| \geq \frac{\epsilon_R}{2}.$$

Using Proposition 10.6, we can exclude the second term in the equation above. Therefore,

$$e_2^{i,j} \implies \left( \hat{\mu}^i_{k^*_{\mathcal{M}(j)}} - \mu_{i,k^*_{\mathcal{M}(j)}} \geq (\mu_{j,k^*_{\mathcal{M}(j)}} - \mu_{i,k^*_{\mathcal{M}(j)}}) - \frac{\eta}{2} - \frac{\epsilon_R}{2} \right).$$

Hence, we conclude that

$$e_2 \implies \hat{\mu}^i_{k^*_{\mathcal{M}(j)}} \geq \mu_{\mathcal{M}(j),k^*_{\mathcal{M}(j)}} - \frac{\eta}{2} - \frac{\epsilon_R}{2}.$$

$\square$

**Claim** 5. Assume that arm $k^*_{\mathcal{M}(j)}$ gets eliminated after round $r$ for user $i$. Then, there must exist an arm $k' \in [K] \setminus k^*_{\mathcal{M}(j)}$ such that

$$\hat{\mu}^i_{k'} - \mu_{\mathcal{M}(i),k'} \geq \epsilon_r - \frac{\eta}{2} - \frac{\epsilon_R}{2}$$

at the end of round $r$.

*Proof.* The arm $k^*_{\mathcal{M}(j)}$ is eliminated after round $r$ only if there exists an arm $k' \in [K] \setminus k^*_{\mathcal{M}(j)}$ such that

$$\hat{\mu}^i_{k'} \geq \hat{\mu}^i_{k^*_{\mathcal{M}(j)}} + \epsilon_r.$$

From Claim 4, we have

$$\hat{\mu}^i_{k^*_{\mathcal{M}(j)}} \geq \mu_{\mathcal{M}(j),k^*_{\mathcal{M}(j)}} - \frac{\eta}{2} - \frac{\epsilon_R}{2}.$$

Substituting this into the previous inequality:

$$\hat{\mu}^i_{k'} \geq \mu_{\mathcal{M}(j),k^*_{\mathcal{M}(j)}} + \epsilon_r - \frac{\eta}{2} - \frac{\epsilon_R}{2}.$$

Since $\mu_{\mathcal{M}(j),k_{\mathcal{M}(j)}^*} \geq \mu_{\mathcal{M}(i),k_{\mathcal{M}(i)}^*}$, we obtain:

$$\hat{\mu}_{k'}^i \geq \mu_{\mathcal{M}(i),k_{\mathcal{M}(i)}^*} + \epsilon_r - \frac{\eta}{2} - \frac{\epsilon_R}{2}.$$

Thus,

$$\hat{\mu}_{k'}^i - \mu_{\mathcal{M}(i),k'} \geq \epsilon_r - \frac{\eta}{2} - \frac{\epsilon_R}{2}.$$

This completes the proof of Claim 5. $\qquad\qquad\square$

*From claim 4 and 5, we conclude that $e_2$ can only occur if:*

- *There exists an agent $i$ and an arm $k_{\mathcal{M}(j)}^* \in \{k_1^*, k_2^*, ..., k_M^*\} \setminus k_{\mathcal{M}(i)}^*$ where $\mu_{\mathcal{M}(j),k_{\mathcal{M}(j)}^*} \geq \mu_{\mathcal{M}(i),k_{\mathcal{M}(i)}^*}$, such that,*

$$\hat{\mu}_{k_{\mathcal{M}(j)}^*}^i \geq \mu_{\mathcal{M}(j),k_{\mathcal{M}(j)}^*} - \frac{\eta}{2} - \frac{\epsilon_R}{2} \tag{12}$$

- *If $k_{\mathcal{M}(j)}^*$ is eliminated after round $r$ from user $i$, then for some arm $k'$,*

$$\hat{\mu}_{k'}^i - \mu_{\mathcal{M}(i),k_{\mathcal{M}(i)}^*} \geq \epsilon_r - \frac{\eta}{2} - \frac{\epsilon_R}{2} \tag{13}$$

*Given $\bar{e}_0$, for Eq. 12 to hold, $k_{\mathcal{M}(j)}^*$ must get eliminated after some round $r \in [1, R]$, as $\frac{\eta}{2} - \frac{\epsilon_R}{2} > \frac{\epsilon_R}{2}$. After $r$ rounds of successive elimination, any arm in the active set will have $\dfrac{8\log\left(\frac{4K_i^2}{\gamma}\right)}{\epsilon_r^2}$ pulls. Hence, from Hoeffding's inequality, the probability that after the $r^{th}$ round,*

$$\mathcal{P}\left(\hat{\mu}_{k_{\mathcal{M}(j)}^*}^i \geq \mu_{\mathcal{M}(j),k_{\mathcal{M}(j)}^*} - \frac{\eta}{2} - \frac{\epsilon_R}{2}\right) = \mathcal{P}\left(\hat{\mu}_{k_{\mathcal{M}(j)}^*}^i \geq \mu_{\mathcal{M}(i),k_{\mathcal{M}(j)}^*} + \left(\mu_{\mathcal{M}(j),k_{\mathcal{M}(j)}^*} - \mu_{\mathcal{M}(i),k_{\mathcal{M}(j)}^*}\right) - \frac{\eta}{2} - \frac{\epsilon_R}{2}\right)$$

$$\leq \mathcal{P}\left(\hat{\mu}_{k_{\mathcal{M}(j)}^*}^i \geq \mu_{\mathcal{M}(i),k_{\mathcal{M}(j)}^*} + \left(\mu_{\mathcal{M}(i),k_{\mathcal{M}(i)}^*} - \mu_{\mathcal{M}(i),k_{\mathcal{M}(j)}^*}\right) - \frac{\eta}{2} - \frac{\epsilon_R}{2}\right)$$

$$\leq \mathcal{P}\left(\hat{\mu}_{k_{\mathcal{M}(j)}^*}^i \geq \mu_{\mathcal{M}(i),k_{\mathcal{M}(j)}^*} + \eta - \frac{\eta}{2} - \frac{\epsilon_R}{2}\right)$$

$$\leq e^{-\frac{4\log\left(\frac{3K_i^2}{\gamma}\right)\left(\frac{\eta}{2} - \frac{\epsilon_R}{2}\right)^2}{\epsilon_r^2}}. \tag{14}$$

*Also, the probability that there exists an arm $k' \in [K] \setminus k_{\mathcal{M}(j)}^*$ such that*

$$\hat{\mu}_{k'}^i - \mu_{\mathcal{M}(i),k_{\mathcal{M}(i)}^*} \geq \epsilon_r - \frac{\eta}{2} - \frac{\epsilon_R}{2} \tag{15}$$

*is less than*

$$(K-1) \cdot e^{-\frac{4\log\left(\frac{3K_i^2}{\gamma}\right)\left(\epsilon_r - \frac{\epsilon_R}{2} - \frac{\eta}{2}\right)^2}{\epsilon_r^2}}.$$

*The events in equations 14 and 15 are independent; hence, we can multiply their corresponding probabilities to obtain the upper bound on $e_2 \cap \bar{e}_0$ as follows:*

$$\implies P(e_2 \cap \bar{e}_0) \leq (M-1).N.(K-1)\sum_{r=1}^{R} e^{-\left(\frac{4\log(\frac{3Ki^2}{\gamma})(\epsilon_r - \frac{\epsilon_R}{2} - \frac{\eta}{2})^2}{\epsilon_r^2} + \frac{4\log(\frac{3Ki^2}{\gamma})(\eta/2 - \frac{\epsilon_R}{2})^2}{\epsilon_r^2}\right)}$$

$$\leq (M-1).N.K.\sum_{r=1}^{R}\left(\frac{\gamma}{3Ki^2}\right)^{4\frac{(\epsilon_r - \frac{\epsilon_R}{2} - \frac{\eta}{2})^2 + (\eta/2 - \frac{\epsilon_R}{2})^2}{\epsilon_r^2}}$$

$\epsilon_R = \eta/17$ *Hence,*

$$\leq (M-1) \cdot N \cdot K \cdot \sum_{r=1}^{R} \left( \frac{\gamma}{3Ki^2} \right)^{\frac{(2\epsilon_r - \frac{18\eta}{17})^2 + \left( \frac{16\eta}{17} \right)^2}{\epsilon_r^2}}$$

$$\leq (M-1) \cdot N \cdot K \cdot \sum_{r=1}^{R} \left( \frac{\gamma}{3Ki^2} \right)$$

$$\leq (M-1) \cdot N \cdot K \cdot \left( \frac{\gamma}{K} \right)$$

**Proposition 10.9.** *From Proposition 1, 2, and 3, the total probability of error until clustering is bounded by:*

$$\mathcal{P}(e_{clustering}) = \mathcal{P}(e_0 \cup e_1 \cup e_2) = \mathcal{P}(e_0 \cup (e_1 \cap \bar{e}_0) \cup (e_2 \cap \bar{e}_0))$$

$$\leq N \cdot K \left( 2\sqrt{2} \cdot \frac{\gamma^{0.75}}{K^{0.75}} + \frac{\gamma}{K} \right) + N\gamma + (M-1) \cdot N \cdot \gamma$$

$$\leq 6NK\gamma^{0.75} \leq \frac{\delta}{2}$$

**Proposition 10.10.** *From Theorem 10.5 and the union bound, the probability of error in the second phase is bounded by:*

$$P(e_{second\_phase}) \leq M \cdot \left( \frac{\delta}{2M} \right) = \frac{\delta}{2} \tag{16}$$

From proposition 10.9 and 10.10 we can get to the statement of theorem 4.3.

### 10.4. Proof of Theorem 2

Assuming our algorithm doesn't enter the error event $e_0$, using equation 3 of Theorem 10.5, for the first phase, we can bound the sample complexity for any arm $i \in [K]$ of any agent by:

$$\leq \sum_{r=1}^{\left\lceil \log_2 \frac{1}{\max(\Delta_i, \eta)} \right\rceil} \frac{8 \cdot \log\left( \frac{4nr^2}{\gamma} \right)}{2^{-2r}}$$

$$\lesssim \max(\Delta_i, \eta)^{-2} \left( \log n + \log\frac{1}{\gamma} + \log\log(\max(\Delta_i, \eta)^{-1}) \right)$$

$$\leq \max(\Delta_i, \eta)^{-2} \left( \log K + \log\delta + \log N + \log\log(\max(\Delta_i, \eta)^{-1}) \right)$$

Hence, we can bound the total sample complexity in the first phase as:

$$T_1 \lesssim \sum_{j \in [N]} \sum_{i=1}^{K} \max\{\Delta_{\mathcal{M}(j),i}, \eta\}^{-2} \cdot \left( \log K + \log\log\left( \max\{\Delta_{\mathcal{M}(j),i}, \eta\}^{-1} \right) + \log N + \log\left( \frac{1}{\delta} \right) \right)$$

Similarly, we can bound the total number of pulls in the second phase as:

$$T_2 \leq \sum_{j \in [C]} \sum_{i=2}^{K} \Delta_{\mathcal{M}(j),i}^{-2} \cdot \left( \log K + \log\log\left( \frac{1}{\Delta_{\mathcal{M}(j),i}} \right) + \log M + \log\left( \frac{1}{\delta} \right) \right)$$

### 10.5. proof of Theorem 5.2

**Proposition 10.11.** *With probability at least $1 - 2\delta/3$, at the end of first phase we will correctly detect*

- *The best arm for all the agents not in set $A$*

- *The best arm for all the $M$ bandits*

*Proof.*

*Claim* 6. Assuming each agent is assigned to one of the bandit randomly with equal probability, then the number of agents picked in first phase is less than or equal to $\frac{\log(\frac{3\cdot M}{\delta})}{\log(\frac{M}{M-1})}$ with probability at least $1-\delta/3$.

*Proof.* The probability by which an agent gets picked from a given bandit will be $1/M$. Hence, the probability that after picking $\frac{\log(\frac{3\cdot M}{\delta})}{\log(\frac{M}{M-1})}$ agents, no agent gets picked learning bandit $m$, is $(1-\frac{1}{M})^{\frac{\log(\frac{3\cdot M}{\delta})}{\log(\frac{M}{M-1})}}$. Putting union bound for all $M$ bandits we arrive at eqn. 17.

$$\mathcal{P}(|[N]\backslash A| > \frac{\log(\frac{3\cdot M}{\delta})}{\log(\frac{M}{M-1})}) \leq M.(1-\frac{1}{M})^{\frac{\log(\frac{3\cdot M}{\delta})}{\log(\frac{M}{M-1})}} \tag{17}$$

$$\mathcal{P}(|[N]\backslash A| > \frac{\log(\frac{3\cdot M}{\delta})}{\log(\frac{M}{M-1})}) \leq \delta/3 \tag{18}$$

$\square$

If successive elimination doesn't enter an error event, then it is clear that the statements of this proposition will be true. Hence, we bound the probability of successive elimination entering an error event for any agent in the first phase.

From the union bound, $\mathcal{P}(e) \leq$ (Number of agents picked in the first round) $\cdot$ (Probability of error for each agent). Hence,

$$
\begin{aligned}
P(e) &\leq \sum_{i\in[N]\backslash A} p(\text{SE giving error for one agent}) \\
&\leq \sum_{i\in[N]\backslash A} \frac{\delta\cdot\log\left(\frac{M}{M-1}\right)}{3\log\left(\frac{3\cdot M}{\delta}\right)} \\
&\leq |[N]\backslash A| \cdot \frac{\delta\cdot\log\left(\frac{M}{M-1}\right)}{3\log\left(\frac{3\cdot M}{\delta}\right)} + P\left(|[N]\backslash A| > \frac{\log\left(\frac{3\cdot M}{\delta}\right)}{\log\left(\frac{M}{M-1}\right)}\right)\cdot 1 \\
&\leq \frac{\log\left(\frac{3\cdot M}{\delta}\right)}{\log\left(\frac{M}{M-1}\right)} \cdot \frac{\delta\cdot\log\left(\frac{M}{M-1}\right)}{3\log\left(\frac{3\cdot M}{\delta}\right)} + \frac{\delta}{3} \\
&\leq \frac{\delta}{3} + \frac{\delta}{3} \leq \frac{2\delta}{3}
\end{aligned}
$$

From above equations and theorem 10.5 we can come directly to the first two statements of the theorem, $\square$

**Proposition 10.12.** *In the second phase, with probability $1 - \frac{\delta}{3}$, we will correctly detect the best arm for all the agents remaining in the set $A$.*

*Proof.* Using the union bound, the probability of error is $\leq N \cdot \frac{\delta}{3N} \leq \frac{\delta}{3}$. Hence, from theorem 10.5, we will detect the best arm with probability at least $1 - \frac{\delta}{3}$. $\square$

**From propositions 10.11 and 10.12, the total probability of error for the $BAI-Cl$ algorithm is less than $\delta$.**

## 10.6. Proof of theorem 5.3

**Proposition 10.13.** *From equation 3, we can say that with probability at least $1 - \frac{2\delta}{3}$, the total number of pulls for the algorithm in the first phase will be less than:*

$$
T_1 \lesssim \sum_{m=1}^{M} \sum_{i=2}^{n} \Delta_{m,i}^{-2} \cdot \left( \log K + \log \gamma + \log\log \Delta_{m,i}^{-1} \right)
$$
$$
+ M \cdot \log\left( \frac{3M}{\delta} \right) \cdot \max_{m=1}^{M} \left( \sum_{i=2}^{n} \max(\eta, \Delta_{m,i})^{-2} \left( \log K + \log \gamma + \log\log\left( \max(\eta, \Delta_{m,i})^{-1} \right) \right) \right),
$$
$$
\gamma = \delta \cdot \frac{\log\left( \frac{M}{M-1} \right)}{\log\left( \frac{3M}{\delta} \right)}
\tag{19}
$$

**Proposition 10.14.** *Similarly with probability at least $1 - \delta/3$ total Number of pulls for the algorithm in the second phase will be less than:*

$$
\leq N.(M.\eta^{-2}(\log M + \log\delta^{-1} + \log N + \log\log\eta^{-1}))
\tag{20}
$$

## 10.7. Proof of theorem 10.1

**Proposition 10.15.** *Given an instance $\nu = ([N], [M], [K], \mathcal{M}, \Pi)$ satisfying assumptions 6.1 and 2.1 with parameters $\eta$ and $\eta_1$, and given that $|\bar{\mu}_j - \mu_{\mathcal{M}(j), k^*_{\mathcal{M}(j)}}| \leq \frac{\eta_1}{4}, \forall j \in S$ from the first phase, the procedure $\widehat{SE}$ identifies the best arm with at least $1 - \gamma$ probability.*

*Proof.* Denote $\hat{a}$ as our potential candidate for the best arm and $a^*$ as the true best arm for the current agent. $\bar{\mu}_{\hat{a}}$ is the estimated mean of arm $\hat{a} = a^*$, which we calculated in the first phase. Therefore, after the $k^{\text{th}}$ round, an error can occur if:

$$
\hat{a} = a^* \quad \text{and} \quad |\hat{\mu}_{\hat{a}} - \bar{\mu}_{\hat{a}}| \geq \frac{\eta_1}{4}, \quad \text{or} \quad \hat{a} \neq a^* \quad \text{and} \quad |\hat{\mu}_{\hat{a}} - \bar{\mu}_{\hat{a}}| \leq \frac{\eta_1}{4}.
$$

Hence:

$$
P(e) \leq \sum_{k=1}^{\infty} \mathbb{P}\left[ \left( \{|\hat{\mu}_{\hat{a}} - \bar{\mu}_{\hat{a}}| \geq \frac{\eta_1}{2}\} \cap \{\hat{a} = a^*\} \right) \cup \left( \{|\hat{\mu}_{\hat{a}} - \bar{\mu}_{\hat{a}}| \leq \frac{\eta_1}{2}\} \cap \{\hat{a} \neq a^*\} \right) \right]
$$

If $\hat{a} = a^*$, then $|\bar{\mu}_{\hat{a}} - \mu_{\hat{a}}| \leq \frac{\eta_1}{4}$, otherwise $|\bar{\mu}_{\hat{a}} - \mu_{\hat{a}}| \geq \frac{3\eta_1}{4}$, hence:

$$
P(e) \leq \sum_{k=1}^{\infty} \mathbb{P}\left[ \left( \{|\hat{\mu}_{\hat{a}} - \mu_{\hat{a}}| \geq \frac{\eta_1}{4}\} \cap \{\hat{a} = a^*\} \right) \cup \left( \{|\hat{\mu}_{\hat{a}} - \mu_{\hat{a}}| \geq \frac{\eta_1}{4}\} \cap \{\hat{a} \neq a^*\} \right) \right]
$$

Using Hoeffding's inequality for 1-sub-Gaussian random variables, we have:

$$
P(e) \leq \frac{2\gamma}{4} \sum_{k=1}^{\infty} k^{-2} \leq \gamma
$$

$\square$

**Proposition 10.16.** *With probability at least $1 - 2\delta/3$, at the end of first phase we will correctly detect*

- *The best arm for all the agents not in set A*

- *The best arm for all the M bandits*

*Proof.* Proof will be same as in 10.11 as we don't change anything in BAI-Cl++ in first phase except calculating additional estimates of the means of the best arms. $\square$

**Proposition 10.17.** *After the first phase, following will hold true.*

$$\mathbb{P}(|\bar{\mu}_j - \mu_j| \leq \eta_1/4, \forall j \in S) \geq 1 - \delta/6$$

*Proof.* We will pull each arm is the set $S$ at-least $\frac{32\log(12M/\delta)}{\eta_1^2}$ times. Hence using union bound over all the arms in the set $S(|S| = M)$ and Hoeffding's inequality for 1-SubGaussian random variable we have,

$$\mathbb{P}(|\bar{\mu}_j - \mu_j| \leq \eta_1/4, \forall j \in S) \geq 1 - \delta/6$$

$\square$

**Proposition 10.18.** *In second phase, with probability $1 - \delta/6$ we will correctly detect the best arm for all the agents remaining in the set $A$*

*Proof.* Using union bound probability of error is $\leq N \cdot \frac{\delta}{6N} \leq \frac{\delta}{6}$, Hence from theorem 10.5 we will detect the best arm with probability at-least $1 - \delta/6$. $\square$

**From proposition 10.15, 10.16, 10.17, 10.18 total probability of error for the $BAI-Cl++$ algorithm is less than $\delta$.**

### 10.8. Proof of Theorem 10.2

**Proposition 10.19.** *In phase $k$, the probability that the $SE(S, \delta_k, R = \log(1/\eta) + 1)$ subroutine returns the true best arm is at least $1 - \delta_k$.*

*Proof.* Since the minimum arm gap is at least $\eta$, under the good event—which occurs with probability at least $1 - \delta_k$—the algorithm returns the best arm while taking at most $\log(1/\eta) + 1$ rounds. $\square$

**Proposition 10.20.** *Given an instance $\nu = ([N], [M], [K], \mathcal{M}, \Pi)$ satisfying Assumptions 6.1 and 2.1 with parameters $\eta$ and $\eta_1$, and given that*

$$|\bar{\mu}_j - \mu_{\mathcal{M}(j), k^*_{\mathcal{M}(j)}}| \leq \frac{\eta_1}{4}, \quad \forall j \in S,$$

*from the first phase, the procedure $\widehat{SE}$ will take at most*

$$\mathcal{O}\left(M \cdot \eta^{-2}(\log M + \log\log\eta^{-1}) + \eta_1^{-2}\log\gamma\right)$$

*pulls.*

*Proof.* Using eqn. 3 we bound the number of pulls in phase $k$ by:

$$O\left(|S| \cdot \frac{\log\left(\frac{|S|\log\eta^{-1}}{2^{-k}}\right)}{\eta^2}\right) + O\left(\frac{\log(4k^2/\gamma)}{\eta_1^2}\right).$$

Denote $E_k$ as the event that the algorithm has not terminated in stage $k$. If the algorithm has not terminated in stage $k$, then it is not the case that $\hat{a}_k = a^*$ and $|\hat{\mu}_a - \bar{\mu}_S^a| < \eta_1/2$.

By a union bound, the probability that these two events **do not** occur is at most

$$1 - \delta_k - \frac{2\gamma}{4k^2} \leq 1 - \left(\delta_k + \frac{2\gamma}{4}\right) \leq \frac{1}{2}.$$

Finally, we obtain:

$$\mathbb{E}_{\nu, \widehat{SE}}[T] \leq \sum_{k=1}^{\infty} \mathbb{P}(E_{k-1}) \cdot \left\{\frac{32}{\eta^2}\log\left(\frac{4k^2}{\gamma}\right) + \mathbb{E}_{\nu, \widehat{SE}\delta_k}[T]\right\}, \tag{21}$$

$$\mathbb{E}_{\nu, \text{Alg}}[T] \lesssim \sum_{k=1}^{\infty} 2^{1-k} \cdot \left\{\frac{32}{\eta^2}\log\left(\frac{4k^2}{\gamma}\right) + M \cdot \frac{\log\left(\frac{M\log\eta^{-1}}{2^{-k}}\right)}{\eta^2}\right\}, \tag{22}$$

$$\mathbb{E}_{\nu, \text{Alg}}[T] \lesssim M \cdot \eta^{-2}(\log M + \log\log\eta^{-1}) + \eta_1^{-2}\log\gamma^{-1}. \tag{23}$$

□

**Proposition 10.21.** *No. of pulls in the first phase of the* `BAI-Cl++` *Algorithm will be less than*

$$T_1 \lesssim \sum_{m=1}^{M} \sum_{i=1}^{K} \Delta_{m,i}^{-2} (\log K + \log \gamma + \log\log \Delta_{m,i}^{-1}) + M.\log\left(\frac{3.M}{\delta}\right). \max_{m \in [M]} \left\{ \sum_{i=1}^{K} \max(\eta, \Delta_{m,i})^{-2} (\log K + \log \gamma \right.$$

$$+ \log\log\max(\eta, \Delta_{m,i})^{-1}) \} + M \left( \frac{\log(\delta^{-1}) + \log(M)}{\eta_1^2} \right), \text{ where } \gamma = \delta. \frac{\log(\frac{M}{M-1})}{\log(\frac{3.M}{\delta})}$$

*Proof.* In the first phase of `BAI-Cl++` we only require an additional $M \cdot \left( \frac{\log(\delta^{-1}) + \log(M)}{\eta_1^2} \right)$ pulls compared to `BAI-Cl`, Hence from eqn. 19 we can conclude the proposition. □

**Proposition 10.22.** *The number of pulls in the second phase of the* `BAI-Cl++` *algorithm will be at most*

$$T_2 \lesssim N \cdot M \cdot \eta^{-2} (\log M + \log\log \eta^{-1}) + N \cdot \eta_1^{-2} (\log \delta^{-1} + \log N). \tag{24}$$

*Proof.* From Proposition 10.20 and setting $\gamma = \delta/6N$, we directly obtain the given claim. □

Adding the number of pulls from Propositions 10.21 and 10.22 completes our proof.

### 10.9. Proof of Theorem 7.1

We start with the first bound. Consider an alternate instance $\nu' \in \mathcal{I}$ which is identical to $\nu$, except that for some $m \in [M]$ and $k > M$, the mean reward for arm $k$ in bandit $m$ is changed to $\mu + 2\eta$. Note that the set of best arms under $\nu$ and $\nu'$ are distinct; in particular, they are $[M]$ and $([M] \setminus \{m\}) \cup \{k\}$ respectively. Hence, any feasible algorithm should be able to reliably infer if the underlying instance is $\nu$ or $\nu'$. Then from (30, Lemma 1) based on a 'change of measure' technique, we have the following lower bound on the expected number of total pulls of arm $k$ by agents learning bandit $m$ under instance $\nu$:

$$\mathbb{E}[T_{m,k}^{\nu}(\mathcal{A})] \geq \frac{\log(1/2.4\delta)}{D(\mu, \mu+2\eta)} = \frac{\log(1/2.4\delta)}{4\eta^2}$$

where $D(a,b)$ denotes the Kullback-Leibler divergence between two Gaussian distributions with means $a$ and $b$ respectively, and is equal to $(a-b)^2$. Summing over all possible alternate best arms $k$ and bandits $m$, we get the first lower bound

$$\mathbb{E}[T_{\delta}^{\nu}(\mathcal{A})] \geq M \cdot (K-M) \cdot \frac{\log(1/4\delta)}{4\eta^2}. \tag{25}$$

We now demonstrate the second lower bound in the expression of the theorem. Again, consider the instance $\nu \in \mathcal{I}$ as defined before. Here, for each agent $i$, we lower bound the total number of samples required from that agent to reliably infer which of the $M$ bandits it is learning. Assume that agent $i$ is learning bandit $m$ (with best arm $m$) under the instance $\nu$; and consider an alternate instance $\nu'$ where it is mapped to a different bandit $m'$ (which by definition has a different best arm $m'$). Note that under either mapping, the mean rewards of all the arms remains the same except arms $m$ and $m'$ for which the mean reward is switched from $\mu + \eta$ to $\mu$ and vice-versa. Clearly, any feasible algorithm should be able to reliably distinguish between the original and alternate problem instances.

Again, using (30, Lemma 1), we have the following lower bound on the expected number of pulls of arms $m$ and $m'$ by agent $i$ under instance $\nu$:

$$\mathbb{E}[T_{i,m}^{\nu}(\mathcal{A}) + T_{i,m'}^{\nu}(\mathcal{A})] \geq \frac{\log(1/2.4\delta)}{D(\mu, \mu+\eta)} = \frac{\log(1/2.4\delta)}{\eta^2}$$

which also serves as a lower bound on the expected number of pulls by agent $i$. Since each agent is independently mapped to a bandit, the total number of pulls across all agents has to satisfy the following lower bound:

$$\mathbb{E}[T_{\delta}^{\nu}(\mathcal{A})] \geq N \cdot \frac{\log(1/2.4\delta)}{\eta^2}. \tag{26}$$

Combining (25) and (26) completes the proof.

### 10.10. Instance Dependent Lower Bound

We can write the following bound on the total expected no. of pulls for any $\delta - PAC$ algorithm for any instance $\nu$ satisfying assumption 1.

$$\mathbb{E}[T^\nu] \geq \max\left( \sum_{m \in M} \sum_{k \in S_{m,\eta}} \frac{\log(1/2.4\delta)}{D(\mu_{\mathcal{M}(i),k}, \mu_{\mathcal{M}(j),k})}, \sum_{i \in [N]} \min_{k \in [K]} \max_{j \in \{j | \mathcal{M}(j) \neq \mathcal{M}(i)\}} \frac{\log(1/2.4\delta)}{D(\mu_{\mathcal{M}(i),k}, \mu_{\mathcal{M}(j),k})} \right)$$

Similar to the proof in theorem 7.1, we will show two lower bounds one for each sub-task, first we show lower bound on identifying for each agent the index of bandit problem that it is learning. Consider an instance $\nu = ([N], [M], [K], \mathcal{M}, \Pi)$ in the set of feasible instances(satisfying assumption 1). Now, assume agent $i$ is learning bandit $m$ in the instance $\nu$; consider an alternate instance where it gets mapped to bandit $m' \neq m$, any correct algorithm should be able to distinguish that for all $m' \neq m$. using (**?** )Lemma 1]kaufmann2016complexity, we can write eq. 27

$$\sum_{k \in [K]} \mathbb{E}[T^\nu_{i,k}].D(\mu_{\mathcal{M}(i),k}, \mu_{\mathcal{M}(j),k}) \geq \log(1/2.4\delta)$$
$$\forall j \in \{j | \mathcal{M}(j) \neq \mathcal{M}(i)\} \tag{27}$$

We can further modify eq.27 to bound total no. of pulls for an agent $i$ as in the following eqn.

$$\mathbb{E}[T^\nu_i] \geq \min_{k \in [K]} \max_{j \in \{j | \mathcal{M}(j) \neq \mathcal{M}(i)\}} \frac{\log(1/2.4\delta)}{D(\mu_{\mathcal{M}(i),k}, \mu_{\mathcal{M}(j),k})}$$

hence, total no. of pulls for a problem instance can be bounded as follows.

$$\mathbb{E}[T^\nu] \geq \sum_{i \in [N]} \mathbb{E}[T^\nu_i] \tag{28}$$

Next, we show a lower bound on the no. of pulls for identifying the best arm of each bandit. Consider the instance $\nu = ([N], [M], [K], \mathcal{M}, \Pi)$ with clustering parameter equal to $\eta$. now for a bandit $m$ consider the set of arms $S_{m,\eta}$ which are at-least $\eta$ worse than the best arm in all the other bandits i.e

$$S_{m,\eta} = \{k | \mu_{m',k} \leq \mu_{m',k^*_{m'}} - \eta, \forall m' \in [M] \backslash m\}$$

We can bound the no. of pulls for a bandit $m$ by change of measure technique, as we for all $m$ in $[M]$ for all arm $k$ in $S_{m,\eta}$ we can consider an alternate instance $\nu'$ where we alter the mean of $k^{th}$ arm to $\mu_{m,k^*_m} + \epsilon$ where $\epsilon$ can be arbitrarily small. The instance $\nu'$ will be an alternate instance as it has different arm for bandit $m$ and it will satisfy the assumption 6.1. Hence, the expected no. of pulls for an arm $k$ of a bandit $m$ is bounded by,

$$\mathbb{E}[T^\nu_{m,k}].D(\mu_{\mathcal{M}(i),k}, \mu_{\mathcal{M}(j),k}) \geq \log(1/2.4\delta), \forall m \in [M] \forall k \in S_{m,\eta}$$

Further, total no. of pulls can be bounded by,

$$\mathbb{E}[T^\nu] \geq \sum_{m \in M} \sum_{k \in S_{m,\eta}} \frac{\log(1/2.4\delta)}{D(\mu_{\mathcal{M}(i),k}, \mu_{\mathcal{M}(j),k})} \tag{29}$$

### 10.11. Additional numerical results

#### 10.11.1. YELP DATASET

We perform experiments using the *Yelp*[6] dataset, which contains ratings for various businesses given by users across different states of the US. We consider $M = 4$ states with the highest number of ratings as our bandits, namely Louisiana, Tennessee, Missouri, and Indiana. We identify $K = 211$ businesses which are present in all the selected states, and these constitute the bandit

---

[6]https://www.yelp.com/dataset

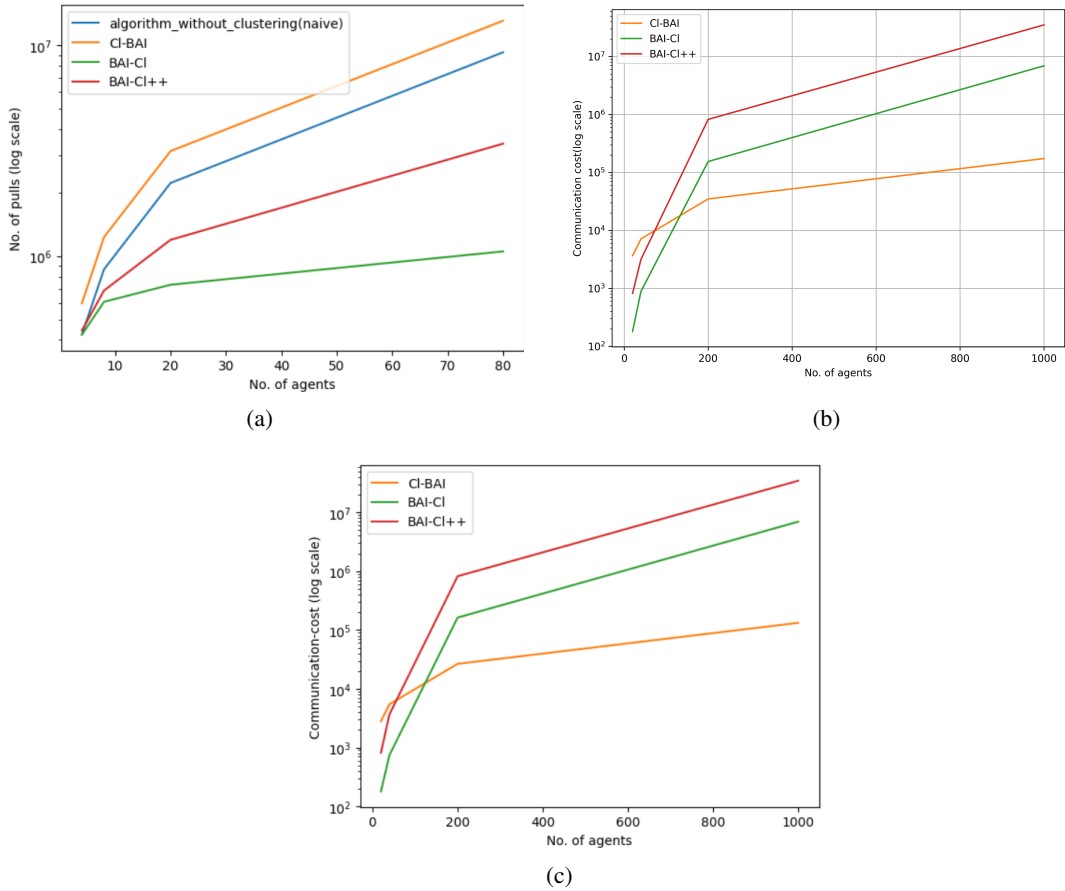

*Figure 2.* (a) Performance with varying number of agents $N$ for Yelp dataset
(b)(c) Communication cost with varying number of agents $N$ for datasets $2, 3$

arms. For each bandit and arm pair (state-business pair), we assign its expected reward to be the average review score the corresponding business got from users in that state. We assume all reward distributions to be 1-Gaussian with the appropriate means.

We find that each of the 4 bandits (states) has a distinct best arm (business with highest average rating). For example, the highest rated business in Louisiana is 'Painting with a Twist', while it is 'Nothing Bundt Cakes' in Indiana. In fact, the dataset satisfies Assumptions 2.1 and 6.1 with clustering parameters $\eta = 0.375$ and $\eta_1 = 0.166$ respectively.

As before, we assume that there are $N$ agents divided into $M$ clusters, each of size $N/M$, and mapped to one of the $M$ bandits. The goal of the learner is to identify the best arm (highest rated business) for each agent.

Figure 2(a) plots the average sample complexity for the various schemes as we vary $N$. Our results demonstrate that clustering-based methods, especially `BAI-Cl++`, significantly reduce the sample complexity compared to the naive scheme. `BAI-Cl` also achieves competitive performance but is less efficient than `BAI-Cl++`. Also, we see that both the naive scheme and Cl-BAI have poor performance. Again, this is consistent with Remark 4.5 since the clustering parameter $\eta$ and the individual bandit arm reward gaps are close in this case.

### 10.11.2. COMMUNICATION COST

We will assume a cost of $c_b = 1$ unit for communicating each bit, and $c_r = 32 c_b = 32$ units for communicating a real number.

Figure 2(b)(c) plots the the overall communication cost of the various algorithms for datasets 2 and 3 (synthetically generated, as described in Section 8), while varying the number of agents $N$. We can see that the communication cost of `Cl-BAI`

is lower than that of `BAI-Cl`, which is itself a little smaller than for `BAI-Cl++`. Note that this order is opposite of that observed for sample complexity, thus indicating a trade-off between the two quantities. This behaviour is also consistent with our observations in Remarks 4.6 and 5.5.

