# OpenReview forum: "Near Optimal Best Arm Identification for Clustered Bandits"
_ICML.cc/2025/Conference — ICML 2025 poster_

### Official Review · Reviewer_n5GW · 2025-03-11

**Overall Recommendation:** 3

**Summary:**

This paper introduces a Best Arm Identification problem with clustering structures. Specifically, given $N$ agents and $M$ bandits instances (Usually $N>M$), each agent faces one of the bandit instance, but the mapping is unknown. The goal is to identify the best arm for each agent with probability at least $1-\delta$. By making use of the clustering structure among the agents, this paper wishes to avoid repeated best arm identification within the same bandit instance, as the agents facing the same instance share the same best arm. Two algorithms (with an improved one) are proposed, CI-BAI and BAI-CI(++), which cluster the arms before and after the best arm identification process, respectively. Theoretical guarantees on the sample complexity are provided and a minimax lower bound is also provided. Experiments are conducted on synthetic and real datasets.

**Claims And Evidence:**

The authors provide proofs for the theorems in the main paper. I skimmed through the proofs and it looks reasonable to me.

**Essential References Not Discussed:**

The references appear well-selected and appropriate.

**Ethical Review Concerns:**

None.

**Experimental Designs Or Analyses:**

For the synthetic dataset, it serves as a sanity check for the proposed algorithms. The experimental results align well with the remarks in the preceding sections.

For the real dataset, while the experimental results appear reasonable, I am not entirely convinced by the choice of \eta. Assumption 2.1 is crucial throughout the paper, as the value of \eta directly impacts the advantage of the proposed algorithms over the naive approach. However, the choice of $\eta$ (and $\eta_1$) seems highly instance-dependent. For example, in the MovieLens dataset, $\eta = 0.0027$, whereas in the Yelp dataset, it is significantly larger at $\eta = 0.375$. Additionally, the authors verify that these datasets satisfy Assumptions 2.1 and 6.1 using ground truth instance parameters, which are not accessible in real-world scenarios.

To address this concern, I recommend that the authors conduct experiments to evaluate the robustness of the proposed algorithms when Assumption 2.1 does not hold or when $\eta$ is misspecified.

**Methods And Evaluation Criteria:**

This paper makes extensive use of the Successive Elimination algorithm, a commonly used method in Best Arm Identification. It is employed to facilitate agent clustering and to identify the best arm for each bandit instance.

**Other Comments Or Suggestions:**

1. Line 274 right: woth high probability -> with high probability.

**Other Strengths And Weaknesses:**

**Strengths**
1. The problem formulation is well stated.
2. Upper bounds of the proposed algorithms are presented, as well as a minimax lower bound which indicates the upper bound is tight in some parameters under some instances.
3. The paper also discussed the communication efficiency of the proposed methods, indicating a trade-off between communication cost and identification efficiency.


**Weaknesses**:
1. The assumptions are quite strong
	1. It assumes different bandit instances have different best arms, which may not be general enough.
	2. Assumption 2.1 requires the knowledge of $\eta$, which is crucial for the proposed algorithm.
2. While Line 248 claims that CI-BAI will be better than the naive algorithm, it is not observed in the experiments with real datasets.
3. It would be great if the authors can specify the results for the case for non-uniform clusters, as mentioned in Remark 5.6, since the assumption that the agents are uniformly distributed is restrictive in Theorem 5.3.
4. The paper relies heavily on Successive Elimination, which makes it difficult for me to discern the technical novelty of the proposed method.

**Questions For Authors:**

1. This paper assumes the number of bandits $M$ is known at the beginning. Is it possible to remove such knowledge? e.g., in the MovieLens experiment, authors manually classify the users according to the ages and obtain $6$ clusters. Is it possible to remove the knowledge of $6$ and let the algorithm learn this automatically?
2. Can the authors highlight the technical contribution of the proposed method?

**Relation To Broader Scientific Literature:**

This paper falls within the field of Bandit Algorithms, specifically focusing on Best Arm Identification with fixed confidence. The authors assume the presence of $N$ agents and $M$ bandit instances, along with an unknown mapping between them. An effective algorithm should leverage the clustering structure to minimize redundant identification efforts.

**Theoretical Claims:**

I skimmed through the proofs for the theoretical guarantees. They appear reasonable to me. I did not check every detail.

---

> ### Author Rebuttal · Authors · 2025-04-01
>
> **Knowledge of $\eta$:** We run additional experiments to validate the robustness of our schemes. Please refer to our rebuttal of Reviewer UFPL.
>
> **Different bandit instances have Different best arms:** Our objective is to find the best arm for every agent, and so it seems reasonable to *define* the clustering based on best arms. Naturally, we put all the agents in a cluster who possess the same best arm. Furthermore, we are able to at least identify a few real-life datasets where these assumptions do hold.
>
> **$\eta-free algorithm:** We can remove the knowledge of $\eta$ from our learning algorithms completely. We can propose a multi-phase algorithm where we start with a large enough value of $\eta$, and at the beginning of each phase, we reduce it by a factor of 2. After some phases, the value of $\eta$ falls below the actual separation and the algorithms start learning the best arm. If we select exponentially increasing phase length, we can show this multi-phase algorithm will succeed in finding the best arms of all the agents. Rigorously proving the correctness is part of our future plans.
>
> **Cl-BAI vs Naive in real data:** As explained in the same paragraph (line 234-248), Cl-BAI will out-perform the naive algorithm if the `separability' parameter $\eta$ is much bigger than the bandit sub-optimality (arm) gaps $\bar{\Delta}$; {in particular when $\eta \gg \bar{\Delta}$}. While this is the case in synthetic datasets, for the real datasets we chose, it turns out (based on our sub-sampling of data) that the bandit arm gaps are comparable to the cluster separation (e.g for Yelp, $\eta=0.375$ and $\bar{\Delta} = 0.25$) , and hence the sample complexities are comparable.
>
> **Results for non-uniform clusters:** We invoke the results of coupon collector problem with unequal probabilities $p_1,\ldots,p_M$, where $p_i$ is proportional to size of cluster $i$. From [1], the term $\mathcal{O}(M\log M/\delta)$ in Theorem 5.3 will be replaced with the following term:
> $
>  \mathcal{O}[\int_0^\infty [1- \prod_{i=1}^M (1-e^{-p_it})] dt]\log(1/\delta).
> $
>
> **Novelty and reliance on SE** Our goal is to perform both clustering and best arm identification jointly, and SE is a common tool to help in both the jobs. We believe the novelty of this work lies in utilizing SE for both clustering and best arm identification judiciously in a *sample efficient* manner. We choose to run SE with different combinations of the subset of arms, success probability and number of rounds to reduce the sample complexity which is highly non-trivial. The *easy tunability* of SE allows us do do this and obtain *near optimal* performance. To the best of our knowledge, this is not done in the literature yet.
>
> This work may be viewed as the first step towards understanding parameter-free clustering in the Federated setup with the objective of finding best arms for all the agents (not *appropriately defined Global best arm*).
>
> **Knowledge of $M$:** The number of clusters $M$ is not needed to be known for Cl-BAI, where the clusters are created based on a nearest-neighbor graph type construction. For BAI-Cl and BAI-Cl++, knowledge of $M$ is needed since the first phase corresponds to recovering the set of all possible best arms using random sampling. Here, we do not require the exact value, any upper bound will suffice with a small increase in sample complexity.
>
> **Technical Contribution:**  (i) *Successive Elimination (SE):*  We believe the contribution of this work lies in utilizing and tuning SE for both clustering and best arm identification judiciously in a *sample efficient* manner which is highly non-trivial. As a result, we obtain *near optimal* performance in terms of sample complexity. Even though this seems apparently simple, to the best of our knowledge, this is currently absent from the literature.
>
> (ii) *Coupon Collector:* We use ideas from the *Coupon Collector* problem to improve the sample complexity of our proposed algorithm, $BAI-Cl$. Using this with SE, we find at least one agent from each cluster with a set of active arms containing the best arms from all $M$ clusters. In the subsequent phase, we let the rest of the agents play only from the obtained subset. Since the cardinality of this set can be much smaller than the total number of arms, we get an improved sample complexity, which is near optimal.
>
> (iii) *Lower Bound* We also use instance perturbation based technique to obtain lower bounds on the expected sample complexity. We obtain both instance dependent as well as instance independent bounds. The lower bound we obtain (nearly) matches the sample complexity of our proposed algorithm rendering them near optimal. Such a lower bound is also novel for this multi agent best arm identification problem.
>
> (iv) *Experiments* We run experiments on real datasets, Yelp and Movielens dataset.
>
> **Reference** [1]  "Birthday paradox, coupon collectors, caching algorithms and self-organizing search", Philippe et.al, Discrete Applied Math, 1992.

---

> > ### Comment · Reviewer_n5GW · 2025-04-02
> >
> > The empirical results provided partially resolve my concern about the robustness of $\eta$. However, (1) the smallest $\eta$ is only half of the ground-truth for each experiment; (2) the largest $\eta$ in Movielens (0.16) is still smaller than the smallest $\eta$ in Yelp (0.187). Due to these setups, the impact of $\eta$ still requires further investigation. In particular, as it is the lower bound on the true $\eta$ that is required as input, running the algorithm with a much smaller $\eta$ is expected. Under such case, it remains unknown whether the algorithm can beat the **parameter-free baselines** or not.
> >
> > Regarding the halving strategy proposed by the authors, it seems to be a possible approach that leads to the $\eta$-free algorithm. However, the design of the stopping rule and the final sample complexity are still unclear. The cumulative sample complexity for this halving strategy may be worse than the parameter-free baselines. As the authors indicate, an $\eta$-free algorithm is of great interest and can greatly enhance this work from my point of view.
> >
> > As the authors have solved my other concerns, I increase my score for the time being. But the authors are **strongly** suggested to conduct further experiments and include the $\eta$-free algorithm in the paper to make the paper complete.

---

> > > ### Author Response · Authors · 2025-04-05
> > >
> > > We thank the reviewer for their comments and agree that understanding the impact of $\eta$, as well as providing a rigorous description and analysis of a parameter-free algorithm are important, and require further careful investigation. We have extended our numerical evaluations on the Movielens and Yelp datasets to include further smaller values of $\eta$, as was suggested; the sample complexity results are tabled below. We have the following observations:
> > >
> > > 1) The naive scheme requires 9.17567403e+09 and 9.2677691e+06 samples for the Movielens and Yelp datasets respectively. Cl-BAI doesn't improve on these; however BAI-Cl does much better even when the assumed value of $\eta$ is much smaller. Intuitively, this is because the first phase of BAI-CL reduces the active arm set size from $K$ to $M$ which provides significant savings in the second phase.
> > >
> > > 2) Even with much smaller (assumed) values of $\eta$, the sample complexity does not vary too much, and so the results are quite robust in that sense.
> > >
> > >
> > > Movie lens:
> > > | \(\eta\)  | Cl-BAI (No. of Pulls) | Cl-BAI (Error) | BAI-Cl (No. of Pulls) | BAI-Cl (Error) |
> > > |-----------|----------------------|---------------|----------------------|---------------|
> > > | 0.16      | 7.9739185e+08     | 10          | 4.27459816e+08       | 10            |
> > > | 0.08      | 1.59879063e+09       | 0            | 9.29866052e+08     | 8          |
> > > | 0.04      | 3.15616948e+09       | 0            | 6.21368206e+08       | 5         |
> > > | 0.02      | 6.33333662e+09       | 0            | 1.02040616e+09       | 0         |
> > > | 0.01      | 1.26017946e+10       | 0             | 3.44881983e+09       | 0             |
> > > | 0.005     | 1.25831127e+10       | 0             | 3.73592584e+09       | 0             |
> > > | 0.0027    | 1.25965021e+10       | 0             | 7.45895904e+09       | 0             |
> > > | 0.0015    | 1.26132350e+10       | 0             | 7.61669485e+09       | 0             |
> > > | 0.00075    | 1.25863287e+10       | 0             | 7.59916579e+09       | 0             |
> > > | 0.000375    | 1.26012931e+10       | 0             | 7.61174240e+09       | 0             |
> > >
> > >
> > > Yelp:
> > >
> > > | \(\eta\)  | Cl-BAI (No. of Pulls) | Cl-BAI (Error) | BAI-Cl (No. of Pulls) | BAI-Cl (Error) |
> > > |-----------|----------------------|---------------|----------------------|---------------|
> > > | 3         | 10,991,609.8         | 10            | 3,680,129.1         | 9             |
> > > | 1.5       | 13,076,497.2         | 10            | 1,117,523.4         | 0             |
> > > | 0.75      | 13,118,020.2         | 0             | 1,032,964.9         | 0             |
> > > | 0.375     | 13,120,822.0         | 0             | 1,078,600.6         | 0             |
> > > | 0.187     | 13,154,100.6         | 0             | 1,001,680.3         | 0             |
> > > | 0.093     | 13,114,919.4         | 0             | 1,123,845.5         | 0             |
> > > | 0.046     | 13,079,106.6         | 0             | 1,149,175.9         | 0             |

---

### Official Review · Reviewer_eVTA · 2025-03-12

**Overall Recommendation:** 4

**Summary:**

The paper explores the problem of identifying the best arms in a multi-agent bandit setting, where agents form (unknown) cluster-based structures.
To address this challenge, the authors propose two algorithms. The first algorithm, CI-BAI, first clusters the agents and then identifies the best arm for a randomly chosen representative from each cluster. The second algorithm follows the same two phases but in a different order.
Theoretical analyses of both algorithms provide insights into their sample complexity and communication efficiency. Additionally, experimental studies demonstrate the effectiveness of the proposed algorithms.

##  Update After Rebuttal
After reviewing the authors’ response, I have decided to maintain my current score.

**Claims And Evidence:**

Yes.

**Essential References Not Discussed:**

I am not aware of any missing prior works.

**Experimental Designs Or Analyses:**

The design of the experiments conducted on synthetic and real datasets appears sound.

**Methods And Evaluation Criteria:**

Yes. The work is primarily theoretical but includes experiments with a reasonably well-designed evaluation.

**Other Comments Or Suggestions:**

The paper lacks a discussion/conclusion section, and I believe the overall writing could be improved. Additionally, there are some typos, which I highlight below:

* Line 117: It would be clearer to replace $ \Delta_{m, k^{*}_{m}} $

with
$\Delta^{*}_{m}$
, for example, as the current notation is somewhat confusing.
* Remark 5.1: "woth" → "with".
* Line 1225: Possible typo and citation issue.

**Other Strengths And Weaknesses:**

Strengths:

1. The paper addresses a novel problem in best-arm identification (BAI) with clustered bandits.
2. To tackle this problem, the authors propose two main algorithms. The second algorithm, BAI-Cl, presents a particularly interesting approach and demonstrates superior performance both theoretically and experimentally.
3. The paper establishes worst-case and instance-based lower bounds for the problem.
4. The conducted experiments show that the proposed algorithms are effective compared to the naive method and also the results align with some findings in the theoretical part.

Weaknesses:
1. It is unclear why the authors did not use the current state-of-the-art algorithm for BAI to achieve better sample complexity.
2. The paper contains numerous typos and needs refinement. Additionally, a more comprehensive discussion of related work could be provided.

**Questions For Authors:**

1. I did not fully understand Remark 4.1. Since the TAS algorithm is optimal, wouldn’t it be more reasonable to incorporate it into the proposed algorithm? What challenges prevent the direct application of TAS?

2. Are there any other relevant algorithms that could tackle the problem, even sub-optimally? In the experiments, you compared your approach only against a naive method for identifying the best arms. Are there any alternative approaches beyond yours and the naive baseline that could serve as a more competitive comparison?

**Relation To Broader Scientific Literature:**

The key contributions of paper are in direction of online decision-making (bandits) and distributed/federated learning.

**Theoretical Claims:**

I reviewed the flow of some parts of the proofs but cannot confirm the correctness of all of them.

---

> ### Author Rebuttal · Authors · 2025-04-01
>
> **Current state-of-the-art algorithm for Best Arm Identification (BAI):**  We choose Successive Elimination (SE) over other state of the art algorithms like Track And Stop (TAS) or Lower Upper Confidence bound (LUCB) for a number of reasons:
>
> (i) We aim to address the problem of Clustering and BAI jointly, and SE is a common tool to address both of these problems. In other words, SE could seamlessly be adapted to our problem formulation. The same thing can not be said for TAS and LUCB.
>
> (ii) For TAS, in general asymptotic guarantees are known, where as we require sharp non-asymptotic guarantees for proving correctness of our proposed algorithms. Some non-asymptotic guarantees of LUCB is known, however they are not immediately adaptable to our problem formulation. Moreover, SE has $3$ tuning parameters, namely the subset of arms, success probability and the number of rounds. Overall, SE is easier to tune compared to other best arm identification algorithms like TAS.
>
> (iii) We would like to clarify that the order-wise performance (sample complexity) of both SE and TAS are similar, whereas SE may have sub-optimal constants. In this work, we provide guarantees on order-wise sample complexity and hence TAS and SE would yield similar results.
>
> (iv) In terms of experiments, one can use other BAI algorithms. However, TAS is computation heavy as compared to SE. Hence, we have taken SE as a default choice.
>
> **Typos and Related work** We apologize for the typos. We have corrected them in the modified version. We have also added a few relevant and recent papers in the Related Work section and provided comparisons with them.
>
> **The paper lacks a discussion/conclusion section** Thank you for the suggestion. Indeed, we have added a Conclusion section now. Additionally we have corrected the typos, actively worked on the writing, and re-organized a few section (including appendix) to improve readability.
>
> **Why optimal TAS is not used?** Although we have discussed this in the first question, we summarize the comparison with TAS here:
>
> (i) Note that for TAS, in general asymptotic guarantees are known, where as we require sharp non-asymptotic guarantees for proving correctness of our proposed algorithms. Moreover, SE has $3$ tuning parameters, namely the subset of arms, success probability and the number of rounds. Overall, SE is easier to tune compared to other best arm identification algorithms like TAS.
>
> (ii) We would like to clarify that the order-wise performance (sample complexity) of both SE and TAS are similar, whereas SE may have sub-optimal constants. In this work, we provide guarantees on order-wise sample complexity and hence TAS and SE would yield similar results.
>
> (iii) We aim to address the problem of Clustering and Best Arm Identification (BAI) jointly, and SE is a common tool to address both of these problems. In other words, SE could seamlessly be adapted to our problem formulation. The same thing can not be said for TAS.
>
> (iv) Finally, in experiments, one can use other Best arm Identification algorithm like TAS. Hoever, TAS is computation heavy compared to SE. Hence, we have taken SE as a default choice.
>
> **Other relevant algorithms and comparison** There are a lot of works in bandit clustering albeit in a parametric setting. In other words, for linear bandits and contextual bandits (which are parameterized), one can naturally define a clustering based on the underlying unknown parameters. On the other hand, needless to say that there is a rich literature on the Best Arm Identification (BAI) problem without clustering structure. However, we are not aware of any other papers in the intersection of non-parametric bandit clustering and BAI, where our work lies.
>
> Alternatively, there are a few works on the BAI problem for Federated Bandits. Out of these, we consider [1]. Note that if we remove the notion of *Global best arm* from [1] and only focus on recovering *Local best arms*, the proposed algorithm there reduces to the naive algorithm we compare against since they do not consider any underlying clustering among agents. We would like to point out that, since we are interested in the BAI for all the agents, naturally the notion of *Global best arm* makes little sense in our setup.
>
> **Reference:** [1] Almost Cost-Free Communication in Federated Best Arm Identification; Kota Srinivas Reddy, P. N. Karthik, and Vincent Y. F. Tan; AAAI 2023.

---

### Official Review · Reviewer_UFPL · 2025-03-13

**Overall Recommendation:** 3

**Summary:**

This work studies the problem of best arm identification for clustering of multi-armed bandits, where $N$ agents are grouped into $M$ clusters, with each cluster solving a stochastic bandit problem. The goal is to identify the best arm for each agent under a $\delta$-probably correct ($\delta$-PC) framework, while minimizing sample complexity and communication overhead.  The authors propose two algorithms: Clustering then Best Arm Identification (C1-BAI) and Best Arm Identification then Clustering (BAI-C1). They provide $\delta$-PC guarantees for both methods, derive bounds on their sample complexity, and provide a lower bound for the problem class. They also propose a variant of BAI-C1 under additional assumptions, which is (order-wise) minimax optimal when $M$ is small. They also provide experimental results to validate the theoretical findings.

**Claims And Evidence:**

Yes.

**Essential References Not Discussed:**

This work is closely related to clustering of bandits. Though the authors have mentioned some works in this line, I think the related literature should be discussed more. Below are some references (not a complete list):
1. Online Clustering of Contextual Cascading Bandits, AAAI 2018.
2. Improved Algorithm on Online Clustering of Bandits, IJCAI 2019.
3. Federated Online Clustering of Bandits, UAI 2022.
4. Online Clustering of Bandits with Misspecified User Models, NeurIPS 2023.

**Experimental Designs Or Analyses:**

No.

**Methods And Evaluation Criteria:**

Yes.

**Other Comments Or Suggestions:**

No.

**Other Strengths And Weaknesses:**

Strengths:

1. To the best of my knowledge, this is the first work to study the best arm identification in the clustering of MAB setting.
2. The paper is well-written and easy-to-follow.
3. The authors provide algorithms with sample complexity bounds, and prove a lower bound.
4. They also provide experimental results to support the theoretical findings.

Weaknesses:

The main weakness I am concerned about is the assumptions (Assumption 2.1 and Assumption 6.1), which seem to be strong. Additionally, for example, under Assumption 2.1, the algorithms need to know $\eta$ (or a lower bound). In the clustering of linear bandit literature, there is also a similar assumption about the separation gap of the different feature vectors, but it does not need to be known. I am wondering if these assumptions can be relaxed.

**Questions For Authors:**

Please see the weaknesses above.

**Relation To Broader Scientific Literature:**

The work is closely related to the literature of clustering of bandits. The authors study the setting of best arm identification for clustering of MAB, which is not yet studied to the best of my knowledge.

**Theoretical Claims:**

No.

---

> ### Author Rebuttal · Authors · 2025-04-01
>
> **Additional references** We thank the reviewer for these pointers. As suggested by the reviewer, we will aim to do a more extensive review of the related literature on clustering in bandits. We would like to point out that one key difference between most of the literature there (including the papers pointed out above) and our work is in how the clusters are defined. While the literature mainly considers parameterized settings such as linear / contextual bandits where the clusters are based on the unknown user preference vectors, our setting is non-parameterized and the cluster definition is based on the mean reward vectors.
>
> **Assumptions 2.1 and 6.1** Thank you for raising this point. Note that our objective is to find the best arm for every agent
> in the system, and so it seems reasonable to *define* the clustering based on the best arms. Assumptions 2.1 and 6.1 essentially quantify a *separability* condition amongst clusters, based on how the best arm of a given cluster performs in other clusters. These assumptions allow us to provide analytical guarantees on the performance of our proposed algorithms; furthermore, as part of our numerical evaluation, we are able to at least identify a few real-life datasets where these assumptions do hold and our proposed schemes are able to provide significant savings in terms of sample complexity and communication cost.
>
> Having said that, there might be other definitions of separability which might also be suited to our setup. However, we choose this since it aligns well with our overall objective of best arm identification.
>
> **Knowledge of $\eta$** The reviewer's point is well-taken. Our response to this is two-fold:
>
> 1) *Robustness to $\eta$*: We have run additional experiments to illustrate that our algorithms are robust to the choice of $\eta$. For the Movielens and the Yelp datasets, we run our algorithms assuming different values of $\eta$ and have tabulated the results below. Each experiment is repeated 10 times.
>
> We have the following observations on the impact of $\eta$-misspecification on correctness and sample complexity. Firstly, as expected, when the assumed $\eta$ is smaller than the true value, the algorithm recovers the best arms correctly. Surprisingly, the same in fact holds true even with larger values of $\eta$ which illustrates that our algorithms are in fact quite robust to the choice of $\eta$. Again, as expected, the incurred sample complexity grows as the gap between the assumed $\eta$ and the true value increases, and there are errors in recovery when $\eta$ is chosen too large.
>
>
> *Movielens*: $M= 6, N = 120, K = 316 , \eta_{true} = 0.0027$
>
> | $\eta$  | Cl-BAI (No. of Pulls) | Cl-BAI (Error) | BAI-Cl (No. of Pulls) | BAI-Cl (Error) |
> |-----------|----------------------|---------------|----------------------|---------------|
> | 0.16      | 7.9739185e+08     | 10          | 4.27459816e+08       | 10            |
> | 0.08      | 1.59879063e+09       | 0            | 9.29866052e+08     | 8          |
> | 0.04      | 3.15616948e+09       | 0            | 6.21368206e+08       | 5         |
> | 0.02      | 6.33333662e+09       | 0            | 1.02040616e+09       | 0         |
> | 0.01      | 1.26017946e+10       | 0             | 3.44881983e+09       | 0             |
> | 0.005     | 1.25831127e+10       | 0             | 3.73592584e+09       | 0             |
> | 0.0027    | 1.25917968e+10       | 0             | 3.90723456e+09       | 0             |
> | 0.0015    | 1.25652013e+10       | 0             | 3.71104286e+09       | 0             |
>
>
> *Yelp*: $M= 4, N = 80, K = 211 , \eta_{true} = 0.375$
>
> | η     | Cl-BAI #Error | Cl-BAI #Pulls | BAI-Cl #Error | BAI-Cl #Pulls |
> |-------|-------------|---------------|-------------|---------------|
> | 3     | 10          | 10991089.6    | 10          | 4930620.3     |
> | 1.5   | 10          | 13096451.2    | 0           | 973381.7      |
> | 0.75  | 0           | 13062668.6    | 0           | 1403340.3     |
> | 0.375 | 0           | 13071359.4    | 0           | 985600.6      |
> | 0.187 | 0           | 13079278.4    | 0           | 1148393.8     |
>
>
>
> 2) *$\eta$ free algorithm*: Alternatively, we can remove the knowledge of $\eta$ from our learning algorithms completely. We can propose a multi-phase algorithm where we start with a large enough value of $\eta$, and at the beginning of each phase, we halve $\eta$. After some phases, the value of $\eta$ falls below the actual gap and the algorithms start learning the best arm. If we select exponentially increasing phase length, we can show this multi-phase algorithm will succeed in finding the best arms of all the agents. Rigorously proving the correctness is part of our future plans.

---

### Official Review · Reviewer_Dq4J · 2025-03-14

**Overall Recommendation:** 4

**Summary:**

The paper considers the problem of federated fixed-confidence best arm identification, where the agents are assumed to be clustered and the agents of the same cluster share the same bandit instance. The authors propose two algorithms, Cl-BAI (cluster-then-BAI) and BAI-Cl (BAI-then-cluster) and show their sample complexities with high probability. Under an additional assumption, the authors propose BAI-Cl++ with improved sample complexity. Then, the authors provide an in-expectation lower bound for the class of instances with mean reward gap assumption, which implies that BAI-Cl++ is orderwise optimal. The algorithms are tested numerically on various datasets, showing their efficacy.

**Claims And Evidence:**

See my comments on Theoretical Results and Experiments

**Essential References Not Discussed:**

None to my knowledge.

**Experimental Designs Or Analyses:**

The experimental designs seem appropriate. Some minor comments:

1. Error bars missing from all experiments (Figure 1, Figure 2)
2. The legends in Figure 1 is too small... As it seems that the legend is the same across the subfigures, consider separating the legend so that the text is legible...
3. Section 8 states that each algorithm went through multiple independent runs. How many?

**Methods And Evaluation Criteria:**

See Experimental Designs or Analyses

**Other Comments Or Suggestions:**

1. Why are all multiplications written as $a.b$? I've never seen this notation...
2. Line 290 (right column): extra $+$
3. Line 346: $>>$ => $\gg$
4. Line 698: $G_i, G_j$?
5. Line 1225: broken citation
6. Please use \appendix before starting the Appendix, and please consider reorganizing/restructuring the Appendix..
7. Misnumbering in the Appendix: What is Theorem 1? (Sec. 9.3)
8. Not defined notations: what is $D(\cdot, \cdot)$ in line 657?
9. The authors should remind in the Appendix that $\epsilon_r = 2^{-r}$.
10. Why say **proof sketch**?
11. Table of sample complexities as well as communication costs for the three algorithms would help a lot in comparing those.
12. It seems that the number of clusters $M$ does not need to be known in advance. The authors should mention this explicitly, as if $M$ is known, then one could just do 1-dimensional $M$-means clustering.
13. (very minor) Could the authors consider using the author-year format (or something similar) throughout the main text? It's very hard to keep track of the references when they are referred to only by numbers.
14. Conclusion (and preferably Future Work) should be included. Maybe move some of the related work part to the Appendix.

**Other Strengths And Weaknesses:**

**Strengths:**
1. New problem setting not considered before
2. Interesting algorithms, especially BAI-Cl; the idea of using the coupon-collected best arms as input to the SE is quite interesting and to the best of my knowledge, novel
3. Extensive related works
4. Experimental results showing the efficacy

**Weaknesses:**
1. Writing should definitely be improved. The Appendix is especially riddled with unnecessary typos, hindering its readability. See Suggestions for a *partial* list of things that I found. This is one of the main reasons of my score not being higher; I could not check the correctness of the proofs in detail.
2. There was a potential error in the proof, which (even though I didn't check the remainder of the proof for the reason mentioned above) makes me question the rigorous correctness of the overall proof. The algorithm design itself looks solid.

**Questions For Authors:**

1. Can the authors comment on the optimality of the communication cost? I also suggest putting in such discussions in the main text.
2. In Algorithm 1 line 17, is the agent selected randomly from each cluster? I don't think this is critical, but for rigorousness, this should be made precise.
3. Can the authors elaborate on why SE is "*easy-to-tune*"?
4. The algorithms are dependent on the knowledge of $\eta$. What happens if the learner misspecifies $\eta$ by overestimating or grossly underestimating it? Especially for the latter case, the authors mentioned in Remark 4.2 that it doesn't change the theoretical results. Does this mean that the theoretical guarantees with the *same* $\eta$ hold regardless of which $\eta' \leq \eta$ is used? Or is the guarantees holding with $\eta$ replaced with $\eta'$?
5. Going further, one future work the authors could mention is making the algorithm parameter-free. For instance, Ghosh et al. (2023) (ref. [40]) proposed a double trick-type wrapper algorithm whose guarantee does not depend on the problem-specific parameter.
6. Instead of constructing a nearest neighbor graph, how about taking a similar approach to Algorithm 2 of Yun & Proutiere (2016)?
7. Overall, what is the main reason (theoretical intuition) for the performance gap between Cl-BAI and BAI-Cl?
8. The upper bounds are with high probability, but the lower bounds are in-expectation. Any chance of closing this gap, either by adapting high-probability lower bound techniques (multiple hypotheses) of Tsybakov (2009), or different analysis of the algorithms?


Se-Young Yun and Alexadre Proutiere. Optimal Cluster Recovery in the Labeled Stochastic Block Model. NIPS 2016 (https://arxiv.org/abs/1510.05956)

**Relation To Broader Scientific Literature:**

- To the best of my knowledge, it tackles a new problem setting of federated FC-BAI with clustering structure
- Interesting algorithmic idea of running SE with coupon-collected best arm candidates.

**Theoretical Claims:**

I didn't check the whole proof in detail, but I have checked and concurred the correctness of the lower bound proof.


I want the authors to clarify the following issue during the proof of Cl-BAI:

In line 658, it states that $i$ and $j$ will not be assigned to the same cluster if $|\hat{\mu}^i - \hat{\mu}^j| > \eta / 2$. However, as Cl-BAI is constructing a nearest neighbor graph, I don't think this is necessarily the case. It may be that there is a $k$ such that $|\hat{\mu}^i - \hat{\mu}^k| \leq \eta / 2$ and $|\hat{\mu}^j - \hat{\mu}^k| \leq \eta / 2$, yet  $|\hat{\mu}^i - \hat{\mu}^j| > \eta / 2$ (e.g., consider $\hat{\mu}^i = \eta / 2, \hat{\mu}^j = - \eta / 2, \hat{\mu}^k = 0$) But, as $i \sim k \sim j$, $i$ and $j$ are indeed in a same cluster.

---

> ### Author Rebuttal · Authors · 2025-04-01
>
> **Nearest neighbor and pairwise distance:** The claim being discussed here considers the `bad' event that two agents belonging to the same cluster get assigned to different clusters, i.e., there exists an arm in the union of the active sets of these two agents, whose estimated means for agents $i$ and $j$ differ by more than $\eta/2$. Our claim is a high probability result and it implies that while the bad event (including for example the specific case that the reviewer has mentioned) can happen, it will occur with very small probability because the underlying ground truth dictates that for any $i,j,k$ which are in the same cluster, their true mean reward vectors will all be identical and thus, when the SE procedure (as prescribed by our algorithm) is run by each of them in the first (clustering) phase, it is very unlikely that the mean reward estimates will be significantly far apart. The claim essentially provides an upper bound on the probability that the bad event occurs.
>
> **Number of independent runs** 50
>
> **Writing, Typos, Figures** We apologize. We have actively worked on the writing in the revised draft and also improved the figures.
>
> **Knowledge of $M$:** The number of clusters $M$ is not needed to be known for  Cl-BAI, where the clusters are created based on a nearest-neighbor graph type construction. For BAI-Cl and BAI-Cl++, knowledge of $M$ (or an upper bound) is needed since the first phase corresponds to recovering the set of all possible best arms.
>
> **Communication cost:** We believe there is a tradeoff between the sample complexity and the communication cost; we see some evidence of this in our analysis. Any algorithm will incur at least $N\cdot \log M \cdot c_b$ communication cost since the learner has to infer each agent's cluster membership. However, we believe this to be weak and identifying the optimal tradeoff is of interest. We have included a more thorough discussion on this.
>
> **Random selection in Algo 1:** The agent is selected arbitrarily from each cluster. We have clarified this in the revised manuscript.
>
> **SE is easy-to-tune:** We say this because SE provides a general procedure which can be altered using three `knobs': the subset of arms, the target error probability, and the number of rounds. We use different combinations of these parameters in different schemes (as well as stages of the same scheme) to achieve various guarantees on surviving arms, their confidence interval sizes etc.
>
> **Robustness with respect to $\eta$:** The reviewer is correct in stating that if $\eta' \leq \eta$, our sample complexity results hold with $\eta$ replaced by $\eta'$. We have run additional experiments to validate the robustness of our schemes with respect to $\eta$. Please see the response for Reviewer  UFPL.
>
> **Parameter-free algorithm** We can remove the knowledge of $\eta$ from our learning algorithms completely. We can propose a multi-phase algorithm where we start with a large enough value of $\eta$, and at the beginning of each phase, we halve $\eta$. After some phases, the value of $\eta$ falls below the actual gap and the algorithms start learning the best arm. If we select exponentially increasing phase length, we can show this multi-phase algorithm will succeed in finding the best arms of all the agents. Rigorously proving the correctness is part of our future plans.
>
> **Yun $\&$ Proutiere, 16** Yun and Proutiere consider cluster recovery in the Stochastic Block Model, where the feedback structure (edge label) is quite different to our setting. However, we do agree that some similar spectral decomposition based method might be feasible for our setting as well.
>
> **Performance gap between Cl-BAI and BAI-Cl** Cl-BAI clusters the users in the first phase and then employs one agent from each cluster to identify the corresponding best arms. However all $K$ arms remain active throughout, including the first phase where all the agents participate. On the other hand, in Cl-BAI, the first phase reduces the active set of arms from $K$ to only $M$ using participation from only $O(M\log M)$ agents and this provides a great reduction in the sample complexity.
>
> **Expected sample complexity** Similar to several works of best arm identification (BAI) in multi-armed bandits, our upper bounds (achievability) are stated as high-probability results whereas the lower bounds are in expectation. However, there are works available in the literature which prove expected sample complexity upper bounds for BAI. For example, Kalyanakrishnan et. al. (2012) for the LUCB scheme and Even-Dar et. al. (2006) for SE style schemes. We believe that similar ideas can potentially be used to derive expected sample complexity bounds for our schemes as well.
>
> **References**
> 1) S. Kalyanakrishnan et. al., PAC subset selection in stochastic multi-armed bandits, ICML 2012.
> 2) E. Even-Dar et. al., Action elimination and stopping conditions for the multi-armed bandit and reinforcement learning problems, JMLR 2006.

---

> > ### Comment · Reviewer_Dq4J · 2025-04-08
> >
> > I thank the authors for their response. I also apologize for the delayed rebuttal comment. Overall, I am satisfied with the authors' responses to my review and the other reviews. One primary concern was the knowledge of $\eta$, which the authors have shown via experimental results that the algorithm is rather robust to misspecification of $\eta$ and that a multiphase, parameter-free algorithm should be possible. In light of this, I raise my score. But, as reviewer n5GW has suggested, **make sure** to include discussions regarding the knowledge of $\eta$ and the parameter-free algorithm that the authors have promised!

---

### Decision · Program_Chairs · 2025-05-01

**Decision:**

Accept (poster)

**Comment:**

This paper consideres the problem of finding the best arm among $K$ arms for each of $N$ agent simultaneously with probability at least $1-\delta$. Each agent belong to one of $M$ groups and reward distribution is per each group. The author proposed algorithm CI-BAI (clustering then BAI) and BAI-CI (BAi then clustering). The sample complexity of the algorithms are analyzed and matches to the possible limit in terms of minimum suboptimality gap $\Delta$ and $K,M,N$.

All reviewers are positive about the paper and these comments are informative enough. The problem setting seems to novel and there is no critical concern, Rev Dq4J considers potential mistake on the proof but it seems the reviewer is fine on the main results. Therefore, I recommends an acceptance. I recommend the authors to revise the paper according to the reviewers' suggestions. For example, most reviewers discuss the knowledge on $\eta$ (suboptimality gap between two clusters). Also Rev n5GW pointed out on the assumption of uniform distribution of the agents to clusters. Finally, please make sure to address the concerns of Rev Dq4J on the solidness of the results.